# Blockwise Flow Matching: Improving Flow Matching Models For Efficient High-Quality Generation

**Dogyun Park**
Korea University
gg933@korea.ac.kr

**Taehoon Lee**
KAIST
dlxogns183@kaist.ac.kr

**Minseok Joo**
Korea University
wlgkcjf87@korea.ac.kr

**Hyunwoo J. Kim**[*]
KAIST
hyunwoojkim@kaist.ac.kr

## Abstract

Recently, Flow Matching models have pushed the boundaries of high-fidelity data generation across a wide range of domains. It typically employs a single large network to learn the entire generative trajectory from noise to data. Despite their effectiveness, this design struggles to capture distinct signal characteristics across timesteps simultaneously and incurs substantial inference costs due to the iterative evaluation of the entire model. To address these limitations, we propose Blockwise Flow Matching (BFM), a novel framework that partitions the generative trajectory into multiple temporal segments, each modeled by smaller but specialized velocity blocks. This blockwise design enables each block to specialize effectively in its designated interval, improving inference efficiency and sample quality. To further enhance generation fidelity, we introduce a Semantic Feature Guidance module that explicitly conditions velocity blocks on semantically rich features aligned with pretrained representations. Additionally, we propose a lightweight Feature Residual Approximation strategy that preserves semantic quality while significantly reducing inference cost. Extensive experiments on ImageNet 256×256 demonstrate that BFM establishes a substantially improved Pareto frontier over existing Flow Matching methods, achieving $2.1\times$ to $4.9\times$ accelerations in inference complexity at comparable generation performance. Code is available at https://github.com/mlvlab/BFM.

## 1 Introduction

Recent advances in generative modeling have been driven by the success of diffusion and flow-matching frameworks [1–4], which learn to transform noise into high-quality data through a sequence of denoising steps. Powered by advanced transformer architectures [5, 6], these approaches have expanded the frontiers of high-fidelity data generation across many fields, including images [7–10], 3D data [11–14] and videos [15–19]. In particular, the Flow Matching (FM) framework [3, 20–22] has emerged as a simple yet effective training paradigm, adopted by several state-of-the-art generative models [23, 24].

A common design choice in FM models is to use a single large neural network to learn the entire velocity field, from noise to data. While parameter-efficient, this monolithic design faces two key limitations. First, the generative trajectory from noise to data inherently involves distinct signal characteristics across time [4]: our frequency-domain analysis (Figure 3) shows that early timesteps

---

[*]Corresponding author.

39th Conference on Neural Information Processing Systems (NeurIPS 2025).

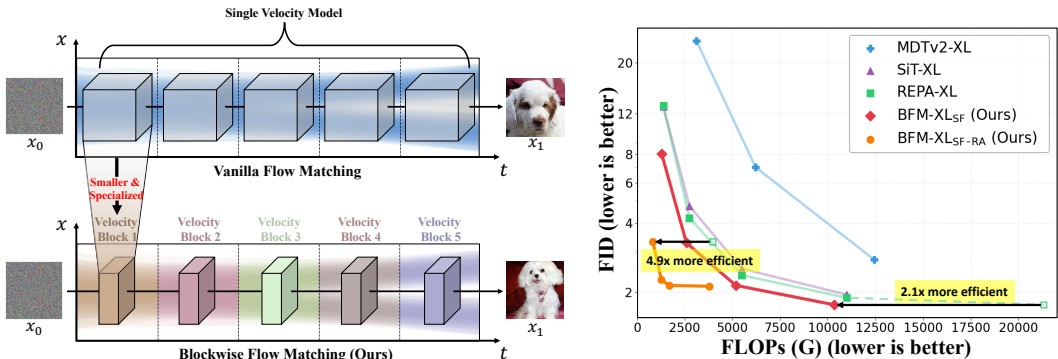

Figure 1: **Blockwise Flow Matching (BFM) achieves a more favorable Pareto-frontier between generation performance (FID) and inference complexity (GFLOPs)** on ImageNet $256 \times 256$. While standard Flow Matching models train a single model, our BFM partitions the flow trajectory into distinct temporal segments, each handled by a **smaller but specialized** velocity block, improving generation quality and efficiency.

are dominated by irregular, low-frequency patterns, while later timesteps require modeling refined, high-frequency details. This temporal heterogeneity imposes conflicting demands on a single shared model, limiting its ability to effectively capture both coarse global structure and subtle local variations simultaneously. Second, using one large model across timesteps incurs high inference costs, as the full model must be evaluated at every solver step, resulting in a total computational complexity of $O(SK)$, where $S$ is the number of steps and $K$ is the model's size in FLOPs. Together, these limitations make it challenging to balance the trade-off between sample quality and inference efficiency.

To address these challenges, we propose *Blockwise Flow Matching* (BFM), a novel framework that rethinks the monolithic architecture of standard FM. Instead of training a *large, shared* model, BFM partitions the flow trajectory into distinct temporal segments, each handled by a *smaller, specialized* velocity block (Figure 1). This segmentation strategy allows each block to focus on a narrower temporal window and better adapt to the specific signal characteristics, as supported by our spectral power analysis (Figure 6). As a result, BFM not only improves generation quality but also reduces inference cost by activating only a subset of the model at each timestep.

However, temporal segmentation limits each block's exposure to clean signals, potentially restricting their capacity to learn high-level semantic features. To mitigate this, we propose a Semantic Feature Guidance (SemFeat), a module that *explicitly conditions* each velocity block on semantic features aligned with powerful pretrained encoders, such as DINOv2 [25] (Figure 2a). SemFeat significantly enhances the generation fidelity, at a cost of increased computation. To reduce this, we propose an additional lightweight module, *Feature Residual Approximation*, which efficiently approximates semantic features during inference, preserving sample quality while significantly reducing computational cost (Figure 2b and 2c).

Through extensive experiments on ImageNet at $256 \times 256$ resolution, we demonstrate that BFM establishes a more favorable Pareto frontier, achieving substantial improvements in both generation quality and inference efficiency over existing state-of-the-art flow matching methods (Figure 1, Table 2). Notably, we achieve comparable or better generation quality with a *2.1× to 4.9× reduction* in inference complexity compared to current best-performing models [26], highlighting the effectiveness of our proposed components.

Our main contributions are summarized as follows:

- We propose Blockwise Flow Matching (BFM), a novel framework that divides the generative trajectory into temporal segments, each modeled by smaller but specialized velocity blocks. This blockwise modeling improves both generation quality and inference efficiency.

- We introduce Semantic Feature Guidance and Feature Residual Approximation modules that significantly enhance visual fidelity while preserving efficiency gains enabled by BFM.

- Extensive experiments demonstrate that BFM achieves a substantially improved Pareto frontier on ImageNet $256 \times 256$, yielding 2.1× to 4.9× faster inference than state-of-the-art generative models with comparable performance.

## 2 Related Work

### 2.1 Diffusion and Flow Matching models

Diffusion models [1, 2, 15, 27, 28] have emerged as powerful tools for generative modeling, achieving state-of-the-art performance across diverse domains [4, 13, 15, 29–31]. More recently, Flow Matching (FM) [3, 20, 21, 32] has been proposed as a general framework that unifies and extends diffusion-based approaches. By directly modeling continuous-time velocity fields between noise and data, FM enables simulation-free training with improved stability and sample quality. Building on this foundation, transformer-based FM architectures like SiT [22] and FiT [6] have achieved strong performance across various generative tasks. However, these models remain computationally expensive at inference time, due to their large model size and the need for repeated evaluations across solver steps. In parallel, there has been growing interest in integrating semantic representation learning into generative models [26, 33, 34]. For instance, RCG [33], REPA [26], and DDT [35] show that semantic features from pretrained visual encoders (e.g., CLIP [36] or DINO [25]) can significantly improve generation fidelity. While these approaches enrich representations, they does not consider distinct characteristics at different diffusion timesteps and are computationally inefficient. In contrast, our method introduces Blockwise Flow Matching with our SemFeat module, enabling efficient and high-quality generation.

### 2.2 Efficient generation

Recent research has explored various strategies to improve the inference efficiency of diffusion and flow-based generative models. One direction involves distillation-based methods, such as consistency models [37–42], distribution matching distillation [43–46], which aim to reduce the number of solver steps by training a student model to imitate the behavior of a pretrained teacher. While effective in accelerating generation, these methods require strong teacher models, multi-stage optimization, and in some cases, adversarial training, which can increase both training complexity and instability. Our method is orthogonal to these approaches and could benefit from distillation techniques to further accelerate generation. Another line of work focuses on model-based compression, which reduces per-step computation by streamlining the network architecture. Prior works such as token merging [47], token pruning [48], and layer pruning [49] propose removing redundant computation during inference. Recently, dynamic inference approaches [50, 51] have introduced input- or timestep-aware mechanisms to adjust the computational graph during generation. However, these methods rely on a single shared model to handle the entire generative trajectory. In contrast, our method partitions the generative trajectory into multiple temporal segments, each modeled by a smaller, specialized velocity block. BFM is complementary to pruning and dynamic routing strategies, which could be applied within individual blocks to further improve efficiency.

## 3 Preliminary

Diffusion and flow-based models [1–3, 20] aim to learn a continuous transformation between a simple reference distribution $\pi_0$ (e.g., Gaussian noise) and a target data distribution $\pi_1$. Given samples $x_0 \sim \pi_0$ and $x_1 \sim \pi_1$, the transformation is defined over a continuous time interval $t \in [0, 1]$ by the following ordinary differential equation:

$$\frac{\mathrm{d}x_t}{\mathrm{d}t} = v(x_t, t), \tag{1}$$

where $x_t$ denotes a time-dependent interpolation between $x_0$ and $x_1$, and $v : \mathbb{R}^d \times [0, 1] \to \mathbb{R}^d$ is the velocity field defined over the data-time joint domain. The interpolation is formulated as:

$$x_t \sim \mathcal{N}(\alpha_t x_1, \sigma_t^2 I) \quad \text{where} \quad \alpha_0 = \sigma_1 = 0, \quad \alpha_1 = \sigma_0 = 1. \tag{2}$$

Different choices of interpolation coefficients $\alpha_t$ and $\sigma_t$ yield different instantiations of diffusion or flow models [20, 52]. Following recent practices [22], we adopt a linear interpolation schedule: $\alpha_t = t$ and $\sigma_t = 1 - t$. To learn the velocity field, a neural network $v_\theta$ is trained to approximate the ground-truth conditional velocity field $v$ by minimizing the mean squared error:

$$\min_\theta \mathbb{E}_{x_0, x_1, t} \left[ \|v(x_t, t) - v_\theta(x_t, t)\|^2 \right]. \tag{3}$$

This formulation is commonly referred to as the standard flow matching framework for training continuous-time generative models.

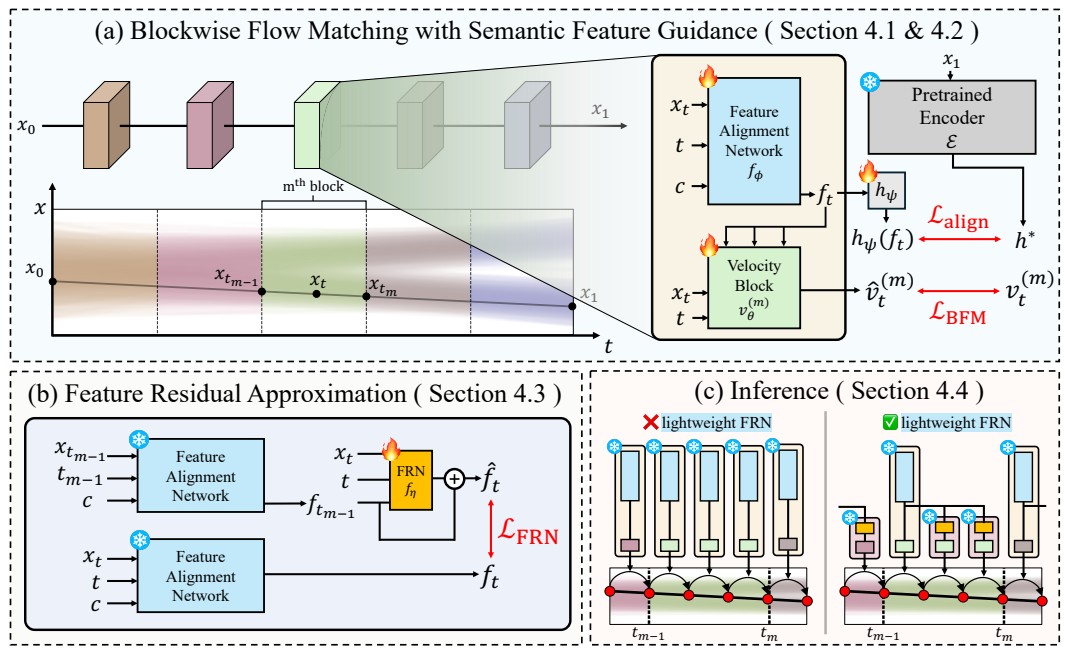

Figure 2: **Overall pipeline of our method.** (a) Blockwise Flow Matching partitions the flow trajectory into $M$ segments, each modeled by a specialized velocity block $v_\theta^{(m)}$. Semantic Feature Guidance enhances the velocity block by explicitly conditioning features $f_t$ from the feature alignment network $f_\phi$. (b) After training velocity models and $f_\phi$, we freeze them and train the Feature Residual Network (FRN) to efficiently approximate $f_t$ by a residual connection. (c) During inference, samples can be efficiently generated by evaluating $f_\phi$ once per segment, reducing inference complexity.

# 4 Method

## 4.1 Blockwise Flow Matching (BFM)

Standard Flow Matching (FM) models [3, 20, 22] train a single large neural network to model the entire flow trajectory from noise to data. However, this design imposes two key limitations. First, it forces a network to simultaneously handle the distinct spectral characteristics at different timesteps [4, 53]. As illustrated in Figure 3, early timesteps are dominated by irregular, low-frequency signals, while later timesteps contain more structured, high-frequency content. This gives conflicting demands on a single model, which may struggle to represent them effectively. Second, it results in high inference cost, as the full model must be evaluated at every solver step, yielding a total complexity of $O(SK)$, where $S$ is the number of solver steps and $K$ is the model's size in FLOPs.

To address these limitations, we propose **Blockwise Flow Matching (BFM)**, which partitions the flow trajectory into a sequence of specialized velocity blocks of model size $K'$,

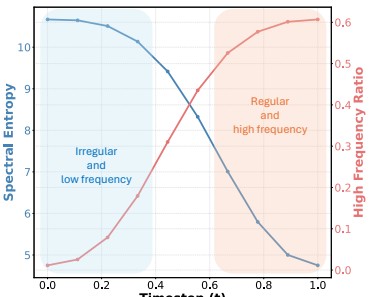

Figure 3: Spectral entropy (SE) and high-frequency ratio across timesteps.

each focusing on a distinct temporal segment. By dedicating each block to a smaller temporal region, these blocks can efficiently capture interval-specific dynamics, allowing the use of smaller networks with size $K' < K$. Consequently, at each timestep, only the corresponding block is evaluated, reducing the inference complexity. Formally, consider flow trajectories defined from pure noise at timestep $t = 0$ to clean data at timestep $t = 1$. We divide this interval into $M$ non-overlapping segments defined by intervals $\{[t_{m-1}, t_m]\}_{m=1}^M$, where $0 = t_0 < t_1 < \cdots < t_M = 1$, and each segment is modeled by its velocity block $v_\theta^{(m)}$. To train the $m$-th block on segment $[t_{m-1}, t_m)$, we

Table 1: **Component analysis**: progressively adding components to our BFM-S backbone. All models are trained for 100K iterations and evaluated using the 250 ODE Euler solver without classifier-free guidance. FLOPs are averaged across solver steps and number of samples. ↓ indicates that lower values are better. Values in parentheses denote relative performance compared to the vanilla FM model (SiT-S). Implementation details are in the supplementary material.

| Method | GFLOPs ↓ | FID ↓ | Solver steps | Avg. Params/step | Total Params |
|---|---|---|---|---|---|
| Vanilla Flow Matching (SiT-S [22]) | 5.45 (×1) | 82.6 | 246 | 33M | 33M |
| **Vanilla BFM-S** ($M = 6$) | 3.64 (×**1.50**) | 81.5 (**−1.1**) | 246 | 18M | 99M |
| + SemFeat (BFM-S$_{SF}$) | 5.01 (×**1.09**) | 66.9 (**−15.7**) | 246 | 23M | 70M |
| + Residual approx. (BFM-S$_{SF-RA}$) | 2.96 (×**1.84**) | 68.3 (**−14.3**) | 246 | 16M | 75M |

define the corresponding start and end points of the segment by interpolating between a clean data sample and a Gaussian noise sample:

$$\text{Start}: \quad x_{t_{m-1}} \sim \mathcal{N}(t_{m-1}x_1, (1 - t_{m-1})^2 I), \tag{4}$$

$$\text{End}: \quad x_{t_m} \sim \mathcal{N}(t_m x_1, (1 - t_m)^2 I). \tag{5}$$

We use sample noise for Equation (4) and (5) to enhance the straightness of the flow trajectory across segments. For a randomly sampled $t \in [t_{m-1}, t_m)$, we sample intermediate data $x_t$ as:

$$x_t = (1 - a_m(t))x_{t_{m-1}} + a_m(t)x_{t_m}, \tag{6}$$

where $a_m(t) = (t - t_{m-1})/(t_m - t_{m-1})$. Then, the ground-truth conditional velocity at time $t$ within the $m$-th segment is given by:

$$v_t^{(m)} = \frac{\mathrm{d}x_t}{\mathrm{d}t} = \frac{x_{t_m} - x_{t_{m-1}}}{t_m - t_{m-1}}. \tag{7}$$

We train each block $v_\theta^{(m)}$ to predict the target velocity using the blockwise flow-matching loss:

$$\mathcal{L}_{\text{BFM}}(\theta) := \mathbb{E}_{x_0,x_1,t}\left[\|v_\theta^{(m)}(x_t, c) - v_t^{(m)}\|^2\right], \tag{8}$$

where $c$ denotes optional conditioning information, e.g., class labels. We omit $t$ for notation simplicity. During inference, given a current state $x_t$ and timestep $t$, we evaluate only the corresponding velocity block $v_\theta^{(m)}$. As empirically demonstrated in Table 1, our proposed BFM-S with six segments (second row) reduces overall inference complexity (FLOPs) while improving generation quality (FID) compared to the standard FM baseline (SiT-S [22]) using the identical transformer blocks.

## 4.2 Semantic Feature Guidance

While our blockwise modeling enables efficient inference and segment-wise specialization, the temporal scope of each velocity block restricts its exposure to the clean data distribution. In particular, blocks operating in early timesteps primarily observe noisy samples, making it challenging to capture high-level semantic features. Motivated by recent studies [26, 33, 34] demonstrating the effectiveness of semantic representations in generative performance, we propose a semantic conditioning technique, called **Semantic Feature Guidance** (SemFeat), which enhances each velocity block by conditioning rich semantic context $f_t$ (Figure 2a).

Specifically, we introduce a *shared* feature alignment network $f_\phi$ which aims to extract robust semantic features from noisy intermediate states $x_t$. To ensure these extracted features carry strong global semantic information, we train this alignment network by aligning its output features with embeddings from pretrained visual encoder $\mathcal{E}$, such as DINOv2 [25]. Given an intermediate noisy input $x_t$ at timestep $t$, the alignment network produces a semantic feature: $f_t = f_\phi(x_t, c)$, where $f_\phi : \mathbb{R}^{d_x} \times [0, 1] \times \mathbb{R}^{d_c} \to \mathbb{R}^{d_x}$ and $c \in \mathbb{R}^{d_c}$ denotes conditioning information. This feature is trained to match the pretrained embedding $h^* = \mathcal{E}(x_1)$ by minimizing the following feature alignment loss:

$$\mathcal{L}_{\text{align}}(\phi, \psi) := \mathbb{E}_{x_0,x_1,t}\left[d\left(h_\psi(f_\phi(x_t, c)), h^*\right)\right], \tag{9}$$

where $h_\psi$ is the learnable projection MLP layer to match the dimensionalities, and $d(\cdot, \cdot)$ is a distance metric such as cosine similarity. Then, the velocity block $v_\theta^{(m)}$ takes the current state

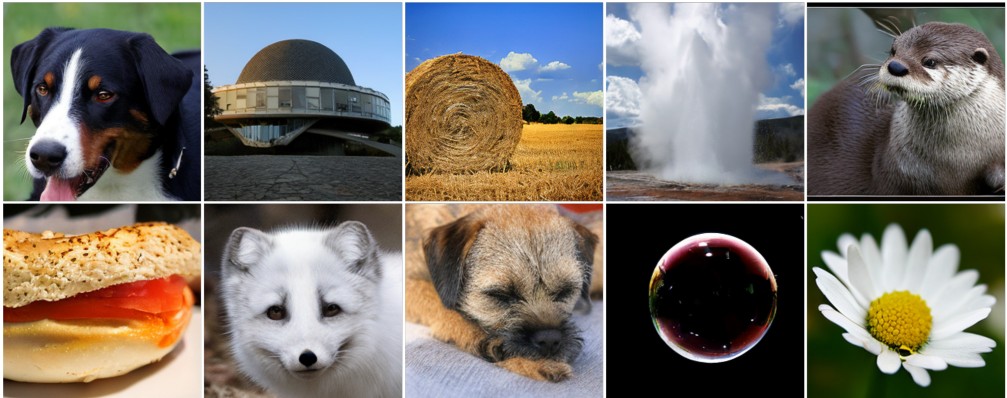

Figure 4: **Samples from BFM-XL$_{\text{SF}}$** on ImageNet $256^2$ with classifier-free guidance ($w = 4.0$).

$x_t$, timestep $t$, condiction $c$, and semantic feature $f_t$ as inputs to estimate the conditional velocity field: $\hat{v}_t^{(m)} = v_\theta^{(m)}(x_t, c, f_t)$. The blockwise flow matching loss with the semantic feature guidance becomes:

$$\mathcal{L}_{\text{BFM}}(\theta, \phi) := \mathbb{E}_{x_0, x_1, t} \left[ \|\hat{v}_t^{(m)} - v_t^{(m)}\|^2 \right]. \tag{10}$$

Finally, the overall training objective is a weighted sum of the blockwise flow matching and feature alignment losses:

$$\mathcal{L} = \mathcal{L}_{\text{BFM}} + \lambda \mathcal{L}_{\text{align}}, \tag{11}$$

where $\lambda$ controls the contribution of the feature alignment objective. Refer to Algorithm 1 for training pseudo code. This conditioning strategy allows each block to access semantically rich guidance, significantly improving generation quality. As demonstrated in Table 1, integrating SemFeat into BFM (third row in Table 1) achieves substantial FID gains compared to the vanilla BFM.

**Comparision with REPA [26].** Our SemFeat is inspired by recent works in representation-aligned diffusion, particularly REPA. However, there is a key difference in design: REPA aligns the internal hidden states of the velocity model directly with semantic features. In contrast, SemFeat introduces a dedicated alignment network $f_\phi$ whose output is explicitly conditioned on the velocity blocks, thereby separating feature alignment from velocity prediction. We observe that this modular design leads to a more coherent and rich representation (Figure 8) and improved generation quality (Table 6).

### 4.3 Feature Residual Approximation

The feature alignment network $f_\phi$ significantly improves generation quality by providing rich semantic guidance. However, evaluating the alignment network at every timestep during inference can undermine the efficiency gains by our BFM. To address this, we propose training a lightweight network that enables an efficient approximation of semantic features during inference (Figure 2b).

Specifically, after the initial training phase where we jointly optimize the feature alignment network $f_\phi$ and the velocity blocks $v_\theta$, we freeze the parameters of both sets of networks. Then, in a second training stage, we introduce a smaller network $f_\eta$, denoted as Feature Residual Network (FRN), to approximate semantic features within the $m$-th temporal segment $[t_{m-1}, t_m]$. This is motivated by the observation in Figure 5 that the discrepancy between start point feature $f_{t_{m-1}}$ and intermediate feature $f_t$ increases as $t$ increases within $[t_{m-1}, t_m]$.

Given the start point $x_{t_{m-1}}$, we first compute its corresponding semantic feature using the alignment network: $f_{t_{m-1}} = f_\phi(x_{t_{m-1}}, c)$. For the subsequent timestep $t$ where $t \in [t_{m-1}, t_m]$, we approximate its feature incrementally via a residual connection defined by: $\hat{f}_t = f_{t_{m-1}} + b_m(t) \cdot f_\eta(x_t, c)$, where $b_m(t) = \frac{t - t_{m-1}}{t_m - t_{m-1}}$ is normalized offset from $t_{m-1}$. This allows the network $f_\eta$ to scale the residual contribution based on how far the timestep $t$ is from the segment start point $t_{m-1}$. FRN $f_\eta$ is trained to match the output of the frozen alignment network $f_\phi$ at intermediate timestep $t$, i.e., $f_t = f_\phi(x_t, c)$, using the following feature regression loss:

$$\mathcal{L}_{\text{FRN}}(\eta) := \mathbb{E} \left[ \|\hat{f}_t - f_t\|^2 \right], \tag{12}$$

Table 2: **System-level comparison** on ImageNet $256 \times 256$ class-conditioned generation with classifier-free guidance [28, 58]. FLOPs are averaged across the solver steps and number of samples. $\downarrow$ and $\uparrow$ indicate whether lower or higher values are better, respectively. $^{\dagger}$ use of guidance scheduling.

| Method | Epochs | #Params. | Step | GFLOPs $\downarrow$ | FID $\downarrow$ | IS $\uparrow$ | Pre. $\uparrow$ | Rec. $\uparrow$ |
|---|---|---|---|---|---|---|---|---|
| ADM [59] | 400 | 554M | 250 | 1120 | 3.94 | 186.7 | 0.82 | 0.52 |
| CDM [27] | 2160 | - | - | - | 4.88 | 158.7 | - | - |
| LDM-4 [4] | 200 | 400M | 250 | 104 | 3.60 | 247.7 | 0.87 | 0.48 |
| VDM++ [60] | 560 | - | - | - | 2.40 | 225.3 | - | - |
| U-ViT-H [61] | 240 | 501M | - | 113 | 2.29 | 263.9 | 0.82 | 0.57 |
| MDTv2-XL [62]$^{\dagger}$ | 1080 | 675M | - | 129.6 | 1.58 | 314.7 | 0.79 | 0.65 |
| MaskDiT [63] | 1600 | 675M | - | - | 2.28 | 276.6 | 0.80 | 0.61 |
| FlowDCN [64] | 400 | 618M | - | - | 2.00 | 263.1 | 0.82 | 0.58 |
| SimpleDiffusion [65] | 800 | 2B | - | - | 2.44 | 256.3 | - | - |
| SD-DiT [34] | 480 | - | - | - | 3.23 | - | - | - |
| FiTv2-3B [6] | 400 | 3B | 250 | 653 | 2.15 | 276.3 | 0.82 | 0.59 |
| FiTv2-XL [6] | 400 | 671M | 250 | 147 | 2.26 | 260.9 | 0.81 | 0.59 |
| DiT-XL [5] | 1400 | 675M | 250 | 114.5 | 2.27 | 278.2 | **0.83** | 0.57 |
| SiT-XL [22] | 1400 | 675M | 250 | 114.5 | 2.06 | 270.3 | 0.82 | 0.59 |
| REPA-XL [26] | 800 | 675M | 250 | 114.5 | 1.80 | 284.0 | 0.81 | 0.61 |
| DiffMoE-L [51] | 600 | 458M | 250 | - | 2.13 | 274.3 | 0.81 | 0.60 |
| DyDiT-XL$_{\lambda=0.7}$ [50] | - | 678M | 250 | 84.3 | 2.12 | 284.3 | 0.81 | 0.60 |
| DyDiT-XL$_{\lambda=0.5}$ [50] | - | 678M | 250 | 57.9 | 2.07 | 248.0 | 0.80 | 0.61 |
| **BFM-XL$_{SF}$ (Ours)** | 400 | 942M | 246 | 107.8 | **1.75** | **289.4** | 0.82 | 0.61 |
| **BFM-XL$_{SF-RA}$ (Ours)** | 600 | 1038M | 246 | **37.8** | 2.03 | 278.1 | 0.80 | 0.62 |

By leveraging this lightweight residual approximation, we drastically reduce inference cost, since the feature alignment network $f_\phi$ is only computed once per segment, and subsequent semantic features within each segment are computed efficiently by the FRN (see Figure 2c). Empirically, our analysis demonstrates that this residual approximation (third row in Table 1) retains over 98% of the BFM-S$_{SF}$ fidelity while reducing inference complexity by 41% (5.01 to 2.96). Refer to Algorithm 2 for pseudo code.

## 4.4 Inference

At the beginning of the $m$-th segment, we compute the semantic feature $f_{t_{m-1}}$ using the feature alignment network $f_\phi$. For subsequent timesteps $t \in [t_{m-1}, t_m)$, we approximate the semantic feature either using FRN $f_\eta$ or the full alignment network $f_\phi$, depending on the desired trade-off between speed and fidelity. Then, the velocity block $v_\theta^{(m)}$ predicts the velocity: $\hat{v}_t^{(m)} = v_\theta^{(m)}(x_t, c, f_t)$. The intermediate state is updated iteratively via a numerical solver using the predicted velocity. This process repeats until reaching $t = 1$, yielding the final sample $x_1 \sim p_{data}$. A full pseudocode implementation of the inference process is provided in the Appendix (Algorithm 3 and 4).

## 5 Experiment

We primarily compare our method with recent flow matching models built on diffusion transformer architectures [5, 6, 22, 26], demonstrating state-of-the-art performance in generative tasks. We follow the experimental setup established by SiT [22] and REPA [26]. Most of the experiments are conducted on the ImageNet dataset at a resolution of $256 \times 256$, unless stated otherwise. We utilize an off-the-shelf pretrained VAE from Stable Diffusion [4], which applies an 8× downsampling factor. We evaluate generation quality using several established metrics, including FID [54], sFID [55], Inception Score (IS) [56], Precision, and Recall [57]. Following recent architectural best practices [5, 6, 22], we adopt a DiT-style transformer backbone for both $v_\theta$, $f_\phi$, and $f_\eta$. To integrate conditions to $v_\theta$, we sum the semantic feature $f_t$ with the timestep embeddings element-wise and inject it at the attention layers via AdaLN-Zero modulation [5]. Our model configurations are based on -S and -XL architectures with a patch size of 2 and $M = 6$. More implementation details are available in the Appendix A.

## 5.1 Comparison with state-of-the-art models

In Table 2, we present a comprehensive comparison of our Blockwise Flow Matching (BFM) framework against recent state-of-the-art generative models. Our BFM with Semantic Feature Guidance (BFM-XL$_{SF}$) achieves an FID score of 1.75, surpassing closely related diffusion transformer models such as FiTv2 [6], DIT [5], SiT [22], and REPA [26]. Notably, compared to the previous best-performing model, REPA, our model demonstrates consistent improvements across FID and IS with a notable improvement of IS by 5.4 points, while also reducing inference complexity in terms of FLOPs by approximately 5% (from 114.5 to 107.8). Furthermore, by incorporating our efficient Feature Residual Approximation module (BFM-XL$_{SF-RA}$), we achieve an even more substantial inference efficiency gain, a 67% reduction in FLOPs over REPA, corresponding to roughly a 3× speed-up. When benchmarked against the recent efficiency-focused model, DyDiT-XL$_{\lambda=0.5}$ [10], our BFM-XL$_{SF-RA}$ not only reduces inference complexity from 57.9 to 37.8 GFLOPs but also significantly improves generation quality, reflected by a 30.1 points increase in IS. These results highlight our model's superior capability in balancing both generation performance and inference efficiency compared to current state-of-the-art approaches.

**Few-step sampling.** We further evaluate our model's performance under a small number of solver steps in Table 3, an essential setting for achieving efficient and practical image generation. In real-world scenarios, reducing the number of function evaluations (NFEs) directly translates to faster sampling and lower inference cost, often at the expense of generation quality. Remarkably, BFM-XL$_{SF}$ achieves an Inception Score of 248.2 and an FID of 8.01 using only **six solver steps**, demonstrating strong robustness with minimal degradation as the number of steps decreases. In contrast, SiT-XL and REPA-XL exhibit

Table 3: Generation performance for different number of solver steps. $\Delta$ indicates the absolute degradation in performance.

| Method | Step | IS $\uparrow$ | $\Delta \downarrow$ | FID $\downarrow$ | $\Delta \downarrow$ |
|---|---|---|---|---|---|
| SiT-XL | 250 | 270.3 | - | 2.06 | |
| | 6 | 179.2 | 91.1 | 12.91 | 10.85 |
| REPA-XL | 250 | 305.7 | - | 1.42 | - |
| | 6 | 182.9 | 122.8 | 13.02 | 11.6 |
| BFM-XL$_{SF}$ | 246 | 315.4 | - | 1.36 | - |
| | 6 | **248.2** | **62.8** | **8.01** | **6.65** |

substantial performance drops of 91.1 and 122.8 in Inception Score, respectively, whereas BFM-XL$_{SF}$ shows a much smaller reduction of 62.8. Furthermore, BFM-XL$_{SF}$ attains an FID of 8.01, outperforming SiT-XL and REPA-XL by large margins of 4.9 and 5.01 FID points, respectively. These results indicate that BFM maintains high-quality generation even under extremely low NFE settings, validating its effectiveness for fast and resource-efficient inference. We attribute this superior few-step performance to the coherent and semantically rich representations provided by our *Semantic Feature Guidance*, which stabilize the velocity field and enable more consistent dynamics even at early timesteps, as visualized in Figure 8.

## 5.2 Generalization and Scalability

**Compatibility to MeanFlow** To further demonstrate the effectiveness of our method, we integrate BFM into the recently proposed MeanFlow [66], which enables high-quality generation using only a few solver steps. Specifically, we adapt our approach by constructing four specialized velocity blocks corresponding to the four-step diffusion trajectory in MeanFlow and applying the MeanFlow objective independently to each temporal segment. This design allows each block to learn distinct dynamics across the generative process, enabling efficient sampling with only **four function evaluations** (one per segment). As summarized in Table 10, our integrated model (BFM$_{SF}$+MeanFlow) achieves a substantially lower FID of 9.7 compared to the MeanFlow baseline of 13.2, despite using the same number of sampling steps. This result highlights the strong compatibility of BFM with existing few-step generative frameworks and demonstrates its effectiveness in improving both efficiency and sample quality with lower inference cost.

**Training at higher resolution.** To examine the scalability of our approach to higher image resolutions, we compare our model against the SiT baseline on ImageNet at $512^2$ resolution. For this setting, we adopt the S-architecture configuration with a patch size of 4. The results, summarized in Table 11, clearly demonstrate that our BFM with Semantic Feature Guidance substantially outperforms the SiT baseline, achieving a remarkable 14.69 improvement in FID. This consistent improvement at a higher resolution highlights the robustness and scalability of our framework in capturing fine-grained visual details and maintaining semantic coherence across larger spatial resolution.

Table 4: Ablation on the number of segments with a fixed number of layers per segment.

| Seg. | Lay. | FLOPs (G) | FID |
|---|---|---|---|
| 4 | 8 | 3.64 | 85.1 |
| 6 | 8 | 3.64 | 81.5 |
| 8 | 8 | 3.64 | 79.1 |
| 12 | 8 | 3.64 | 76.7 |

Table 5: Ablation on the number of segments with a fixed total network capacity.

| Seg. | Lay. | FLOPs (G) | FID |
|---|---|---|---|
| 4 | 12 | 5.46 | 81.7 |
| 6 | 8 | 3.64 | 81.5 |
| 8 | 6 | 2.72 | 88.1 |
| 12 | 4 | 1.82 | 95.2 |

Table 6: Comparison of our proposed SemFeat and REPA [26] on FM and BFM framework.

| Method | FLOPs (G) | FID |
|---|---|---|
| (a) FM + REPA | 5.45 | 72.9 |
| (b) FM + SemFeat | 6.88 | 68.5 |
| (c) BFM + SemFeat | **5.01** | **66.9** |

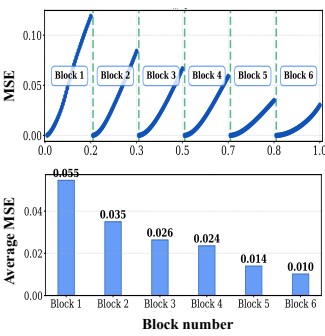

Figure 5: Discrepancy between $f_{t_{m-1}}$ and $f_t$ within each block averaged over 50 samples.

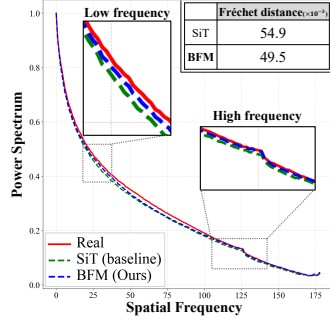

Figure 6: Fourier power spectrum of real images, SiT-generated images, and BFM.

Figure 7: Training loss $\mathcal{L}_{\text{FRN}}$ over time with and without feature residual approximation.

## 5.3 Analysis

To show the efficacy of our components, we conduct a comprehensive component-wise analysis.

**Effect of temporal segments.** First, we investigate how varying the number of temporal segments $M$ in BFM influences generation quality and computational efficiency under two complementary settings. This analysis provides insight into how temporal decomposition affects specialization and scalability within the flow-matching framework.

1) *Fixed per-segment capacity.* In Table 4, we fix the number of layers within each segment while varying the total number of segments $M$, thereby keeping the per-segment capacity constant. We observe that increasing $M$ consistently improves FID scores, suggesting that finer-grained temporal partitioning enables each velocity network to specialize more effectively in modeling localized dynamics along the generative trajectory. Importantly, these improvements come without any increase in inference complexity, highlighting BFM's ability to scale up without added inference cost.

2) *Fixed total network capacity*: In Table 5, we fix the overall model capacity by increasing the number of segments while proportionally decreasing the number of layers per segment. This setup demonstrates a trade-off: while more segments provide greater temporal specialization, shallower blocks may lack sufficient capacity to model complex signals. Empirically, we find that a moderate segmentation level (e.g., $M = 6$) achieves the optimal balance, allowing for effective specialization while preserving sufficient expressiveness within each block.

**Spectral analysis.** To further investigate whether blockwise modeling promotes specialized learning across different stages of the generative trajectory, we conduct a frequency-domain analysis of generated images. Following the methodology of [67], we compute the 2D Fourier power spectrum for both real and generated images and apply azimuthal integration to derive their mean spectral power distributions. This analysis allows us to assess how well each model captures structural details across spatial frequencies, from coarse global layouts to fine-grained textures.

As illustrated in Figure 6, the Fréchet distance between the spectral distributions of real images and those generated by BFM-XL$_{\text{SF}}$ (0.049) is smaller than that of SiT-XL (0.054), indicating that our model more faithfully reproduces the frequency characteristics of natural images across the spectrum. This result supports our core design hypothesis: segment-wise specialization enables different blocks

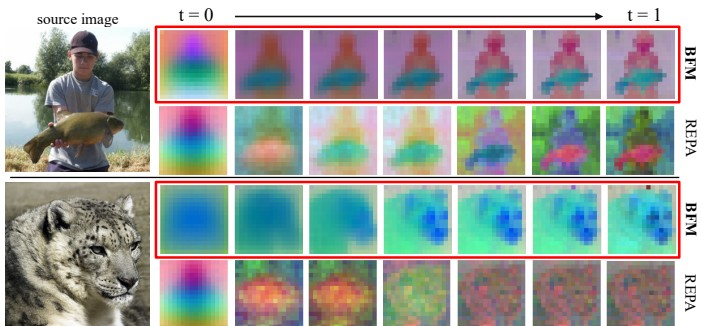
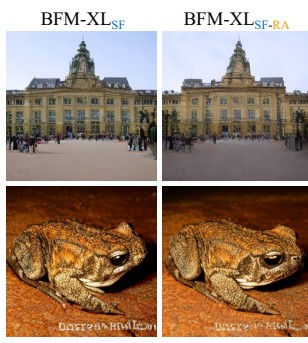

Figure 8: **Semantic features over timesteps.** We visualize the PCA of features from BFM and REPA for the source images. Compared to REPA, the semantic features extracted from BFM are more consistent across timesteps.

Figure 9: Generated images from BFM without (left) and with (right) Residual Approximation.

to more effectively capture diverse signal characteristics across the diffusion trajectory, leading to improved spectral fidelity.

**SemFeat vs. REPA.** We analyze the effectiveness of our proposed SemFeat by comparing it against the closely related approach, REPA [26]. As discussed in Section 4.2, SemFeat differs from REPA in a key design: SemFeat leverages an external feature alignment network that explicitly conditions features on the velocity model, whereas REPA aligns semantic features implicitly within the velocity model itself. This modular design separates representation learning from generative modeling, enabling SemFeat to learn more coherent and stable semantic representations.

To empirically validate this, we conducted a feature-level analysis by applying Principal Component Analysis (PCA) to the semantic features $f_t$ extracted from each model. We visualize the top three components in Figure 8: SemFeat exhibits high semantic consistency across varying noise levels, capturing temporally stable and semantically meaningful information. In contrast, REPA generates noticeably noisier and less consistent semantic features, even within the same object region, highlighting the advantage of SemFeat's modular conditioning. Quantitative comparisons in Table 6 confirm these observations. Starting from standard FM with REPA (a), replacing REPA with SemFeat (b) yields a significant FID improvement of 4.4 points, underscoring the effectiveness of explicit modular conditioning. Moreover, integrating SemFeat within our BFM framework (c) further improves the FID score by an additional 1.6 points, while reducing inference complexity.

**Effectiveness of feature residual approximation.** To demonstrate the effectiveness of our feature residual approximation approach, we first compare it against a direct approximation method that independently predicts semantic features, i.e., $\hat{f}_t = f_\eta(x_t, c)$ without residual connections. Figure 7 shows that our residual approximation method converges faster and achieves lower feature approximation loss ($\mathcal{L}_{\text{FRN}}$), highlighting the efficacy of modeling semantic features as residual increments from the segment's start point. Additionally, as demonstrated in Table 1 and 2, incorporating residual approximation significantly reduces inference complexity: BFM-S$_{\text{SF-RA}}$ reduces 41% FLOPs compared to BFM-S$_{\text{SF}}$ and BFM-XL$_{\text{SF-RA}}$ reduces 65% FLOPs compared to BFM-XL$_{\text{SF}}$, all while preserving generation quality. Qualitative visual comparisons in Figure 9 also demonstrate marginal perceptual differences, validating efficient feature approximation.

## 6 Conclusion

In this paper, we introduced *Blockwise Flow Matching* (BFM), a novel generative framework that partitions the generative trajectory into distinct temporal segments, each handled by compact and specialized velocity blocks. This design allowed each block to effectively capture the unique signal characteristics of its interval, leading to improved generation quality and efficiency. By further incorporating our proposed *Semantic Feature Guidance* and *Feature Residual Approximation* modules, we demonstrated that BFM establishes a significantly improved Pareto-frontier between generation performance and inference complexity on ImageNet 256×256: *2.1× and 4.9× acceleration* in inference complexity compared to the existing state-of-the-art generative methods.

## Acknowledgement

This work was partly supported by the National Supercomputing Center with supercomputing resources including technical support(KSC-2024-CRE-0273, 30%), the Korea Research Institute for defense Technology planning and advancement-Grant funded by Defense Acquisition Program Administration(DAPA) (KRIT-CT-23-021, 30%), the Institute of Information & communications Technology Planning & Evaluation(IITP) grant funded by the Korean government(MSIT) (RS-2024-00457882, AI Research Hub Project, 30%), and (RS-2025-25442149, LG AI Star Talent Development Program for Leading Large-scale Generative AI Models in the Physical AI Domain, 10%).

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

# A Implementation details

Table 7: Architecture and training configurations of Blockwise Flow Matching

| | Vanilla BFM-S (Table 1) | BFM-S$_{SF}$ (Table 1,4,6,7,12) | BFM-S$_{SF-RA}$ (Table 1) | BFM-XL$_{SF}$ (Table 2,3,5,7-10) | BFM-XL$_{SF-RA}$ (Table 2,8,10) |
|---|---|---|---|---|---|
| Num. segments $M$ | 6 | 6 | 6 | 6 | 6 |
| ***Each Velocity Block*** $v_\theta^{(m)}$ | | | | | |
| Num. layers | 8 | 4 | 4 | 5 | 5 |
| Hidden dims | 384 | 384 | 384 | 1152 | 1152 |
| Num. heads | 6 | 6 | 6 | 16 | 16 |
| ***Feature Alignment Network*** $f_\phi$ | | | | | |
| Num. layers | – | 6 | 6 | 20 | 20 |
| Hidden dims | – | 384 | 384 | 1152 | 1152 |
| Num. heads | – | 6 | 6 | 16 | 16 |
| ***Feature Residual Network*** $f_\eta$ | | | | | |
| Num. layers | – | – | 2 | – | 4 |
| Hidden dims | – | – | 384 | – | 1152 |
| Num. heads | – | – | 6 | – | 16 |
| ***Training Config.*** | | | | | |
| Optimizer | AdamW | AdamW | AdamW | AdamW | AdamW |
| Learning rate | 1e-4 | 1e-4 | 1e-4 | 1e-4 | 1e-4 |
| Batch size | 864 | 864 | 864 | 864 | 864 |
| $\lambda$ | – | 0.5 | – | 0.5 | – |
| Visual Encoder $\mathcal{E}$ | – | DINOv2-B | DINOv2-B | DINOv2-B | DINOv2-B |
| $d(\cdot, \cdot)$ | – | cosine sim. | cosine sim. | cosine sim. | cosine sim. |

We provide the overall hyperparameter setup in Table 7.

**Training.** We implement our model based on the original SiT implementation [22] with recent improvements such as SwiGLU [68] activations, RMS normalization [68], and Rotary Positional Embeddings (RoPE) [69]. Following FiTv2 [6], we reassign model parameters and adopt AdaLN-LoRA [70] within transformer blocks to mitigate the parameter overhead of AdaLN modules. This adjustment substantially reduces the total parameter count while maintaining comparable performance.

We use pre-computed latent vectors derived from raw pixels via the Stable Diffusion VAE, without applying data augmentation, following [26]. For feature projection, we employ a three-layer MLP with SiLU activations. We adopt uniform temporal partitioning with $t_m = \frac{m}{M}$ for simplicity, though preliminary experiments indicate that non-uniform, data-adaptive partitioning could further improve performance. To train multiple velocity networks across temporal segments, one simple approach is to randomly sample a velocity network per iteration; however, this may lead to uneven updates among networks. Instead, we uniformly divide each training batch according to the number of temporal segments and assign corresponding timesteps to each subset. For example, with a batch size of 256 and four temporal segments, each segment receives 64 samples. This training strategy ensures balanced updates across all velocity networks and reduces training variance by evenly sampling timesteps across the entire time horizon. All experiments are conducted with a global batch size of 864 using eight NVIDIA A100 GPUs. Full training pseudocode is provided in Algorithms 1 and 2.

**Training complexity.** Our Blockwise Flow Matching (BFM) can be viewed as a form of sparse activation, analogous to a Mixture-of-Experts (MoE) model with a fixed router, where each velocity network serves as an expert. Similar to MoEs, BFM increases the total parameter count while maintaining comparable computational cost and latency. We evaluate the training resource usage on ImageNet $256 \times 256$ for XL configuration using NVIDIA A100 GPU with a batch size of 32. We use deepspeed profiler to measure the FLOPs and results are summarized in Table 8. We use multiply-accumulates (MACs) to represent FLOPs. For both simple and balanced training strategies, BFM achieves per-iteration FLOPs and training throughput comparable to SiT and REPA. Although BFM introduces more parameters, its memory footprint increases only moderately (from 30–31 GB to 47 GB), remaining well within the capacity of modern GPUs. These results demonstrate that BFM is both practical and scalable for large-scale training. Furthermore, it is possible to train each block parallelly (and separately) across multiple GPUs, which can further improve the training scalability.

Table 8: **Training efficiency.** Per-iteration computational cost, throughput, and memory usage. † indicates our balanced training strategy.

| Method | TFLOPs / iter. | Iters / sec | Mem. (GB) |
|---|---|---|---|
| SiT-XL/2 | 7.59 | 2.54 | 30 |
| REPA-XL/2 | 7.72 | 2.62 | 31 |
| **BFM-XL/2 (Ours)** | 7.47 | 2.78 | 45 |
| **BFM-XL/2 (Ours)**† | 7.71 | 2.24 | 47 |

Table 9: **Inference efficiency.** Wall-clock runtime and total FLOPs at $256\times256$ resolution with 246 NFEs.

| Method | NFE | Runtime (s) | GFLOPs |
|---|---|---|---|
| SiT-XL/2 | 246 | 44.51 | 114.5 |
| REPA-XL/2 | 246 | 44.51 | 114.5 |
| **BFM-XL/2$_{\text{SF}}$ (Ours)** | 246 | 40.37 | 107.8 |
| **BFM-XL/2$_{\text{SF-RA}}$ (Ours)** | 246 | 19.42 | 37.8 |

Table 10: Compatibility to MeanFlow [66] framework on ImageNet $256^2$ for 400K training iterations.

| Model | NFE | FID ↓ |
|---|---|---|
| MeanFlow | 4 | 13.2 |
| **BFM$_{\text{SF}}$+MeanFlow** | 4 | **9.7** |

Table 11: System-level performance comparison on ImageNet $512^2$ resolution for 400K training iterations.

| Method | FID ↓ | NFE ↓ |
|---|---|---|
| SiT-S/4 | 102.54 | 246 |
| **BFM-S/4 (Ours)** | **87.85** | 246 |

**Inference.** For inference, we use the Euler solver with the ODE (Tables 1,3,4,5,6 in the main paper) or the Euler-Maruyama solver [22] with the SDE (Table 2 in the main paper). We keep the 246 solver steps (41 solver steps for each segment) except for Table 3 in the main paper.

Specifically, given an initial noise input $x_{t_0} = x_0 \sim p_{\text{noise}}$, the generation proceeds sequentially through each temporal segment $[t_{m-1}, t_m)$. At the beginning of the $m$-th segment, we compute the semantic feature $f_{t_{m-1}}$ using the feature alignment network $f_\phi$. For subsequent timesteps $t \in [t_{m-1}, t_m)$, we approximate the semantic feature either using FRN $f_\eta$ or the full alignment network $f_\phi$, depending on the desired trade-off between speed and fidelity. Then, the velocity block $v_\theta^{(m)}$ predicts the velocity: $\hat{v}_t^{(m)} = v_\theta^{(m)}(x_t, c, f_t)$. The intermediate state is updated iteratively via a numerical solver using the predicted velocity. This process repeats until reaching $t = 1$, yielding the final sample $x_1 \sim p_{\text{data}}$. A full pseudocode implementation of the inference process is provided in Algorithm 3 and 4.

**Inference complexity.** We benchmark the wall-clock inference time of our models against baseline methods on a single NVIDIA A100 GPU with a batch size of 32. The results are summarized in Table 9. Empirically, our primary model, BFM$_{\text{SF}}$, achieves a **9.4% reduction** in inference time than the baselines while attaining better FID scores. Furthermore, the most efficient variant, BFM$_{\text{SF-RA}}$, achieves a remarkable **63% reduction** in runtime while maintaining performance comparable to SiT. These results demonstrate that the reduction in FLOPs achieved by BFM translates directly into real-world speedups, validating its inference efficiency.

**Metrics.** We evaluate generation performance using standard metrics including FID [54] (Fréchet Inception Distance), IS [56] (Inception Score), and Precision/Recall [57, 71]. Unless otherwise noted, we follow the evaluation protocol of [59] and report results using 50K generated samples.

FID is the most commonly used metric, measuring the feature distance between real and generated image distributions. It is computed using the Inception-V3 network under the assumption that both feature distributions follow multivariate Gaussian statistics.

IS, also based on Inception-V3, evaluates both image quality and diversity by measuring the KL-divergence between the marginal and conditional label distributions derived from the network logits.

Finally, Precision captures the fraction of generated samples that lie close to the real data manifold, while Recall measures how well the generated distribution covers real data samples. Together, these complementary metrics provide a more holistic assessment of generative performance in terms of both fidelity and diversity.

## B  Analysis details

**Component analysis (Table 1).** We provide additional implementation details for the experimental configurations reported in Table 1 of main paper. For the vanilla Flow Matching model, we train

---

**Algorithm 1** Training Blockwise Flow Matching (BFM)

---

**Require:** Feature alignment network $f_\phi$, velocity network $v_\theta$, and projection layer $h_\psi$
  1: **repeat**
  2:     Sample a mini-batch of size $B$: $\{(x_0^{(i)}, x_1^{(i)}, c^{(i)})\}_{i=1}^B$          $\triangleright$ $x_0$: data, $x_1$: noise
  3:     Initialize losses: $\mathcal{L}_{\text{BFM}} \leftarrow 0$, $\mathcal{L}_{\text{align}} \leftarrow 0$
  4:     Partition sample indices $i$ uniformly into $M$ disjoint groups: $\mathcal{I}_1, \ldots, \mathcal{I}_M$
  5:     **for** each segment $m = 1, \ldots, M$ **do**
  6:         **for** each sample index $i \in \mathcal{I}_m$ **do**          $\triangleright$ All samples can be computed parallel
  7:             Sample $t^{(i)} \sim \mathcal{U}[t_{m-1}, t_m)$
  8:             $x_{t_{m-1}}^{(i)} \leftarrow (1 - t_{m-1})x_0^{(i)} + t_{m-1}x_1^{(i)}$
  9:             $x_{t_m}^{(i)} \leftarrow (1 - t_m)x_0^{(i)} + t_m x_1^{(i)}$
 10:             $a_m(t)^{(i)} \leftarrow \frac{t^{(i)} - t_{m-1}}{t_m - t_{m-1}}, \quad x_t^{(i)} \leftarrow (1 - a_m(t)^{(i)})x_{t_{m-1}}^{(i)} + a_m(t)^{(i)} x_{t_m}^{(i)}$
 11:             **Semantic features:** $h^{*(i)} \leftarrow \mathcal{E}(x_1^{(i)}), \quad f_t^{(i)} \leftarrow f_\phi(x_t^{(i)}, c^{(i)})$
 12:             **Segment velocity target:** $v_t^{(m,i)} \leftarrow \dfrac{x_{t_m}^{(i)} - x_{t_{m-1}}^{(i)}}{t_m - t_{m-1}}$
 13:             **Per-sample losses:**
 14:                 $\mathcal{L}_{\text{align}} \mathrel{+}= d\left(h_\psi(f_t^{(i)}), h^{*(i)}\right)$
 15:                 $\mathcal{L}_{\text{BFM}} \mathrel{+}= \left\| v_\theta^{(m)}(x_t^{(i)}, c^{(i)} f_t^{(i)}) - v_t^{(m,i)} \right\|^2$
 16:         **end for**
 17:     **end for**
 18:     Normalize losses: $\mathcal{L} \leftarrow \frac{1}{B}\left(\mathcal{L}_{\text{BFM}} + \lambda \mathcal{L}_{\text{align}}\right)$
 19:     Update $\theta, \phi, \psi$ with a gradient step on $\mathcal{L}$
 20: **until** convergence

---

---

**Algorithm 2** Training Feature Residual Network (FRN)

---

  1: Freeze parameters $v_\theta, f_\phi\ h_\psi$, and introduce FRN parameters $f_\eta$
  2: **repeat**
  3:     Sample $m \sim \mathcal{U}[1, ..., M]$
  4:     Sample $t \sim \mathcal{U}[t_{m-1}, t_m)$
  5:     Compute semantic feature at segment start: $f_{t_{m-1}} = f_\phi(x_{t_{m-1}}, c)$
  6:     Compute semantic target at intermediate step $t$: $f_t = f_\phi(x_t, c)$
  7:     Compute normalized time offset $b_m(t) = \frac{t - t_{m-1}}{t_m - t_{m-1}}$
  8:     Approximate semantic feature at intermediate step $t$: $\hat{f}_t = f_{t_{m-1}} + b \cdot f_\eta(x_t, c)$
  9:     Compute residual approximation loss $\mathcal{L}_{\text{FRN}} = \|\hat{f}_t - f_t\|^2$
 10:     Update FRN parameters $\eta$ via gradient step on $\mathcal{L}_{\text{FRN}}$
 11: **until** convergence

---

the original SiT-S model (12 transformer blocks) with architectural improvements. For our vanilla BFM-S model (second row of Table 1), we train the same transformer architecture (except that we reassign model parameters with AdaLN-LoRA) with six temporal segments. We assign each velocity block with 8 transformer blocks. When adding a feature alignment network (BFM-S$_{\text{SF}}$, third row of Table 1), we reduce the number of layers of each velocity block, as shown in Table 7.

**Spectral Entropy and High-Frequency Ratio (Figure 3).** To analyze the frequency characteristics of generated images across timesteps, we compute two key metrics: Spectral Entropy (SE) and the High-Frequency Ratio (HFR). Spectral Entropy quantifies the distributional complexity of a signal in the frequency domain. Specifically, we treat the normalized 2D power spectrum of an image as a probability distribution and compute its Shannon entropy. A higher SE indicates a more uniform, less structured distribution of spectral energy (i.e., more randomness), while a lower SE reflects a concentration of energy in fewer frequencies, indicating more structured signals. High-Frequency

---

**Algorithm 3** Inference

---

1: Initialize: $x_{t_0} = x_0 \sim p_{\text{noise}}$
2: **for** $m = 1, 2, ..., M$ **do**
3:     Define $K$ solver steps within segment: $t_{m-1} = t^{(0)} < \cdots < t^{(K)} = t_m$
4:     **for** $k = 0, 1, ..., K - 1$ **do**
5:         Compute semantic feature: $f_{t^{(k)}} = f_\phi(x_{t^{(k)}}, c)$
6:         Compute velocity: $\hat{v}_{t^{(k)}} = v_\theta^{(m)}(x_{t^{(k)}}, c, f_{t^{(k)}})$
7:         Update state with solver $\Phi$ step size $\Delta t$:
8:             $x_{t^{(k+1)}} = \Phi(x_{t^{(k)}}, \hat{v}_{t^{(k)}}, \Delta t)$
9:     **end for**
10: **end for**
11: Output final generated sample: $x_1 \sim p_{\text{data}}$

---

---

**Algorithm 4** Efficient inference with Feature Residual Network

---

1: Initialize: $x_{t_0} = x_0 \sim p_{\text{noise}}$
2: **for** $m = 1, 2, ..., M$ **do**
3:     Compute semantic feature once at segment start:
4:         $f_{t_{m-1}} = f_\phi(x_{t_{m-1}}, c)$
5:     Define $K$ solver steps within segment: $t_{m-1} = t^{(0)} < \cdots < t^{(K)} = t_m$
6:     **for** $k = 0, 1, ..., K - 1$ **do**
7:         Compute normalized time offset $b_m(t^{(k)}) = \frac{t^{(k)} - t_{m-1}}{t_m - t_{m-1}}$
8:         Approximate semantic feature efficiently: $f_{t^{(k)}} = f_{t_{m-1}} + b_m(t^{(k)}) \cdot f_\eta(x_{t^{(k)}}, c)$
9:         Compute velocity: $\hat{v}_{t^{(k)}} = v_\theta^{(m)}(x_{t^{(k)}}, c, f_{t^{(k)}})$
10:         Update state with solver $\Phi$ step size $\Delta t$:
11:             $x_{t^{(k+1)}} = \Phi(x_{t^{(k)}}, \hat{v}_{t^{(k)}}, \Delta t)$
12:     **end for**
13: **end for**
14: Output final generated sample: $x_1 \sim p_{\text{data}}$

---

Ratio measures the proportion of total spectral energy in the high-frequency range. It captures the relative contribution of fine-grained details in the image.

To compute both metrics, we randomly sample 10,000 real images from the ImageNet set and transform them to noisy versions at specific timesteps by interpolation formulation. Each image is converted to the frequency domain via a 2D Fourier Transform. We then calculate the power spectrum and apply azimuthal integration to obtain the spectral distribution. SE is computed as the Shannon entropy of the normalized spectrum, while HFR is obtained by summing the energy above a predefined frequency threshold (=0.5) and dividing by the total energy.

**Fourier power spectrum (Figure 6).** We perform a frequency-domain analysis to evaluate how well the spectral characteristics of generated images match those of real images. This experiment tests whether our blockwise modeling strategy improves spectral fidelity by allowing different blocks to specialize in capturing distinct frequency components along the generative trajectory. Specifically, we randomly sample 100 images from each model (BFM-XL$_{\text{SF}}$ and SiT-XL [22]) and 100 real images from ImageNet. We compute the 2D Fourier power spectrum for each image and apply azimuthal integration to obtain a 1D mean spectral power distribution, following the procedure in [67]. We then calculate the Fréchet Distance [72] between the spectral distributions of generated and real images to quantify their similarity.

**Discrepancy between $f_{t_{m-1}}$ and $f_t$ (Figure 5).** For each segment $m$, we randomly sample 50 images from the ImageNet dataset and convert them into their corresponding noisy versions $x_{t_{m-1}}$ and $x_t$ at timesteps $t_{m-1}$ and $t \in [t_{m-1}, t_m)$, respectively We then evaluate the temporal consistency of the semantic features extracted by the alignment network by computing the mean squared error (MSE) between $f_{t_{m-1}} = f_\phi(x_{t_{m-1}}, c)$ and $f_t = f_\phi(x_t, c)$. A lower MSE indicates that the feature alignment network learns temporally stable, time-invariant representations across the segment.

**Quantitative comparison between REPA [26] and SemFeat (Table 6).** We provide additional implementation details for the experimental configurations reported in Table 6. All models are trained for 100K iterations using DINOv2-B [25] as the target representation, with a loss weight of $\lambda = 0.5$ and negative cosine similarity as the alignment objective. Feature projection layers are implemented with identical configurations across all methods for a fair comparison. (a): We implement REPA using the SiT-S architecture (12 transformer layers), aligning the hidden representations at the 8th transformer layer to the DINOv2-B targets. (b): We incorporate our proposed SemFeat into the standard FM framework by decomposing the SiT-S model into two modules: a feature alignment network composed of 8 transformer layers, and a velocity model composed of 4 transformer layers. (c): This setting corresponds to our BFM-S$_{\text{SF}}$ in Table 7.

**PCA results (Figure 8, 10)** For the PCA analysis, we extract semantic features from noisy inputs $x_t$ using both REPA-XL and BFM-XL$_{\text{SF}}$. In the case of REPA-XL, we use the hidden representations from the 8th transformer layer, as this layer is aligned with DINO features during training. For BFM-XL$_{\text{SF}}$, we directly use the outputs of the feature alignment network, $f_t = f_\phi(x_t, c)$. We perform Principal Component Analysis across features collected from multiple timesteps to capture their dominant variations. The top three principal components are then mapped to the RGB channels for visualization, allowing us to assess the semantic consistency and structure of features across time.

# C  Additional PCA results

In Figure 10, we present additional PCA visualizations of semantic features across varying noise levels. SemFeat consistently produces semantically coherent and temporally stable representations, maintaining structural consistency even under significant noise. In contrast, REPA exhibits greater variability and noise in its feature representations, with noticeable inconsistencies across the same object regions. These observations further support the effectiveness of SemFeat's modular conditioning in capturing robust and meaningful semantic information throughout the generative trajectory.

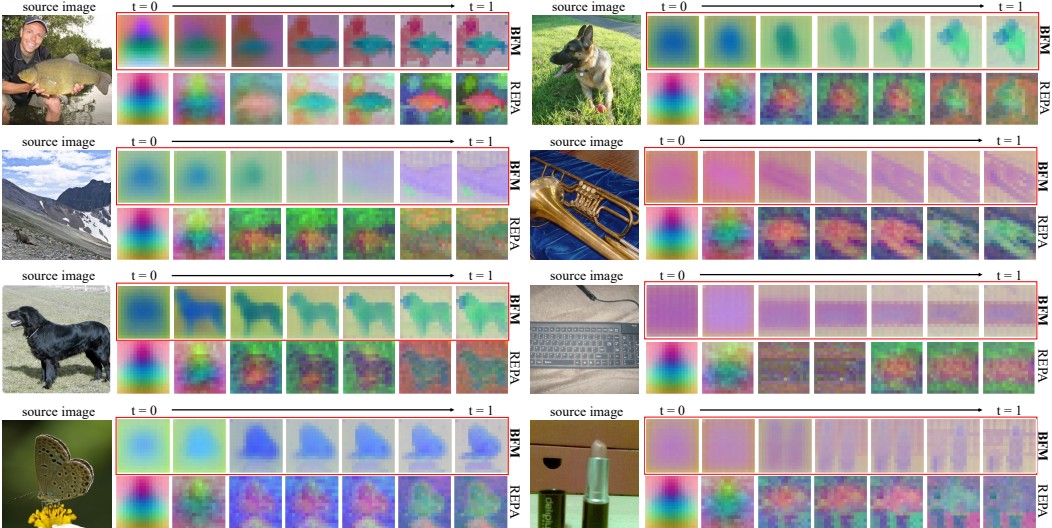

Figure 10: **Semantic features over timesteps**. We visualize the PCA of features from BFM and REPA for the source images. Compared to REPA, the semantic features extracted from BFM are more consistent across timesteps.

# D   More qualitative results

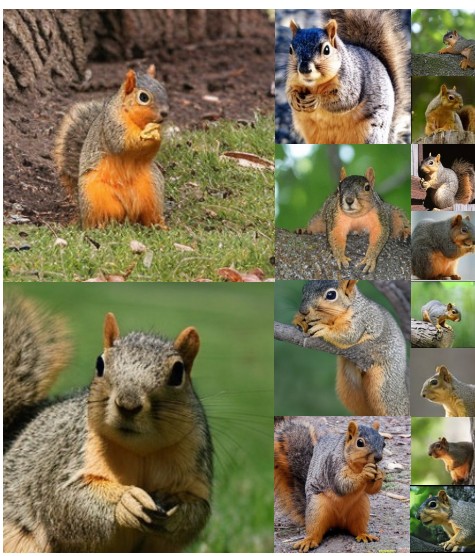

Figure 11: **Uncurated generation results of BFM-XL$_{SF}$**. We use classifier-free guidance with $w = 4.0$. Class label = "fox squirrel" (335).

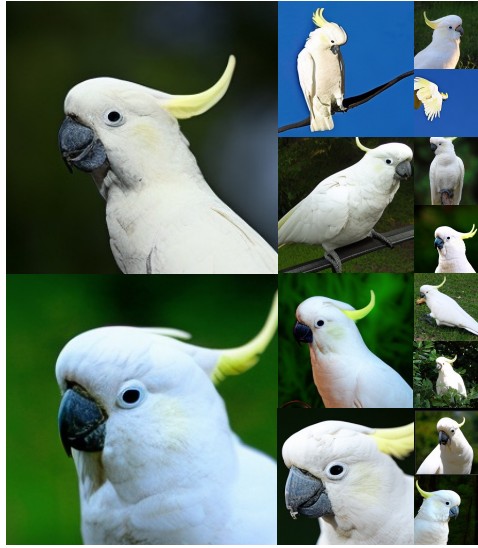

Figure 12: **Uncurated generation results of BFM-XL$_{SF}$**. We use classifier-free guidance with $w = 4.0$. Class label = "Cacatua galerita" (89).

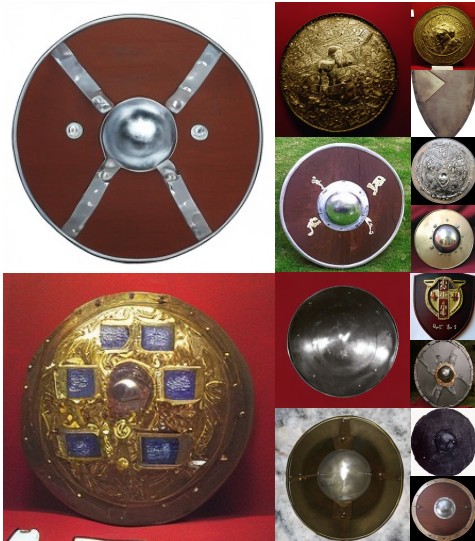

Figure 13: **Uncurated generation results of BFM-XL$_{SF}$**. We use classifier-free guidance with $w = 4.0$. Class label = "shield, buckler" (787).

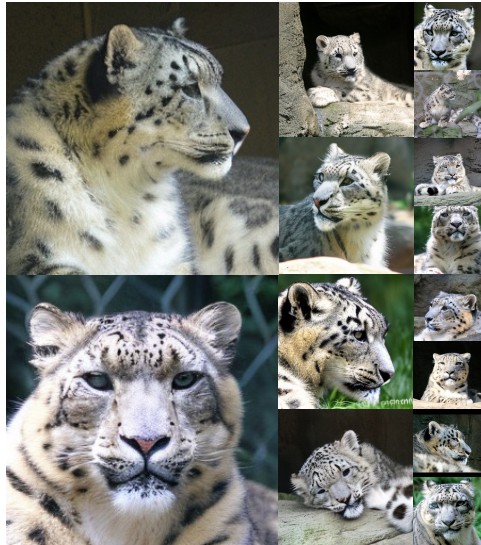

Figure 14: **Uncurated generation results of BFM-XL$_{SF}$**. We use classifier-free guidance with $w = 4.0$. Class label = "snow leopard" (289).

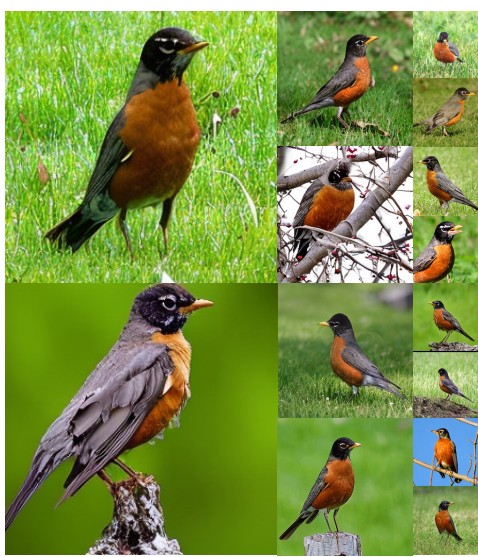

Figure 15: **Uncurated generation results of BFM-XL$_{SF}$**. We use classifier-free guidance with $w = 4.0$. Class label = "American robin" (15).

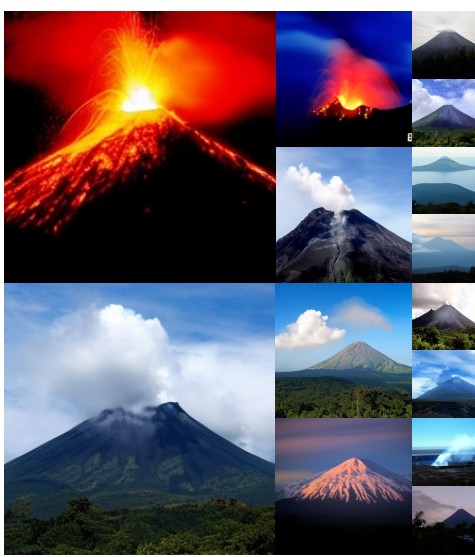

Figure 16: **Uncurated generation results of BFM-XL$_{SF}$**. We use classifier-free guidance with $w = 4.0$. Class label = "volcano" (980).

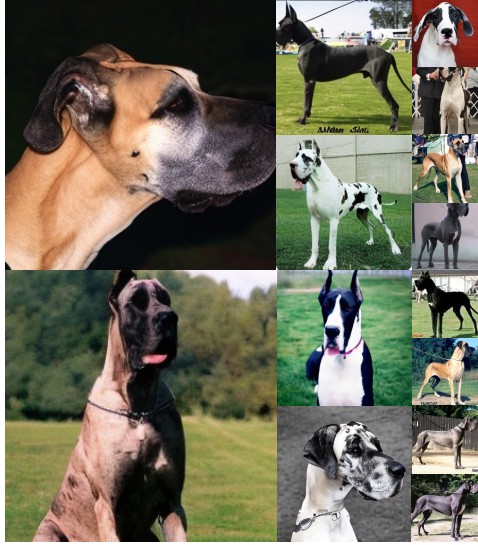

Figure 17: **Uncurated generation results of BFM-XL$_{SF}$**. We use classifier-free guidance with $w = 4.0$. Class label = "Great Dane" (246).

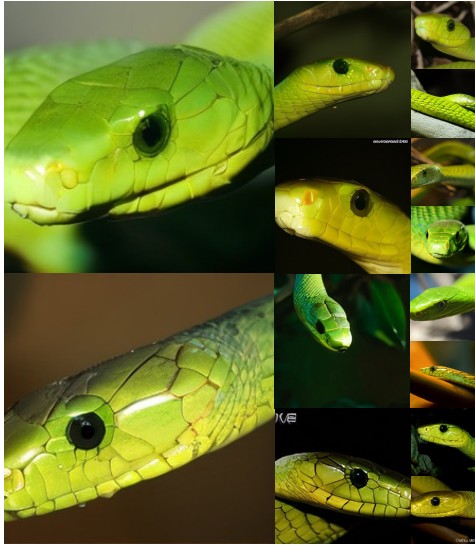

Figure 18: **Uncurated generation results of BFM-XL$_{SF}$**. We use classifier-free guidance with $w = 4.0$. Class label = "green mamba" (64).

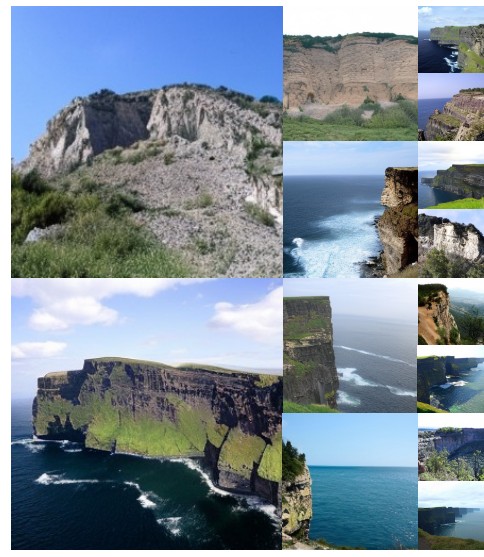

Figure 19: **Uncurated generation results of BFM-XL$_{\text{SF}}$**. We use classifier-free guidance with $w = 4.0$. Class label = "Cliff" (972).

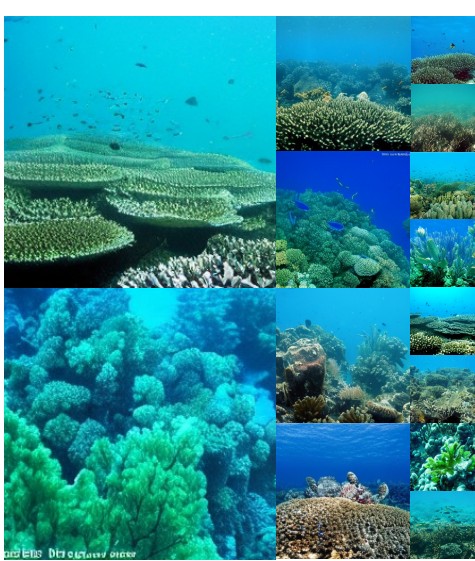

Figure 20: **Uncurated generation results of BFM-XL$_{\text{SF}}$**. We use classifier-free guidance with $w = 4.0$. Class label = "Coral Reef" (973).

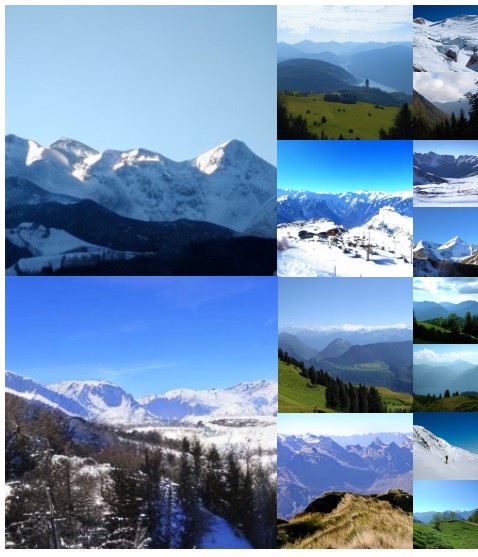

Figure 21: **Uncurated generation results of BFM-XL$_{\text{SF}}$**. We use classifier-free guidance with $w = 4.0$. Class label = "Alp" (970).

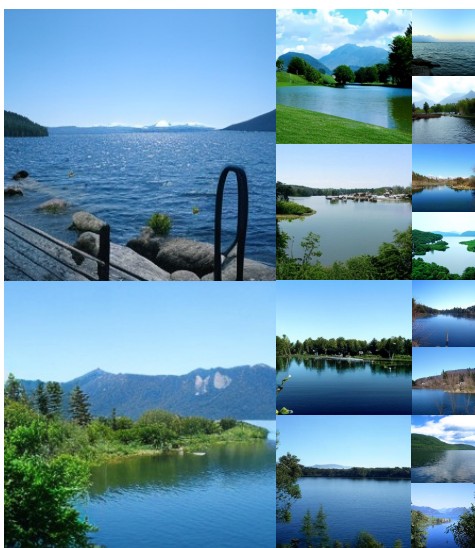

Figure 22: **Uncurated generation results of BFM-XL$_{\text{SF}}$**. We use classifier-free guidance with $w = 4.0$. Class label = "Lakeside" (975).

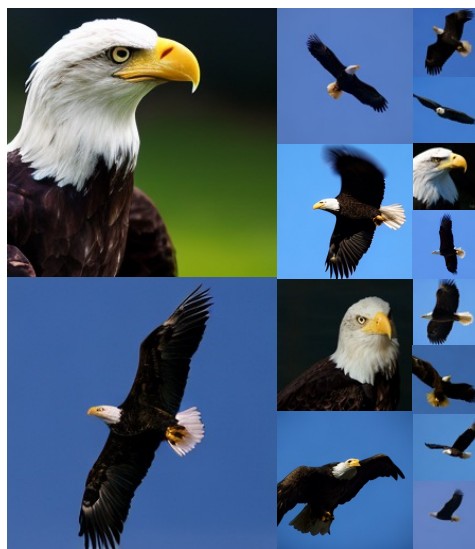

Figure 23: **Uncurated generation results of BFM-XL$_{\text{SF}}$**. We use classifier-free guidance with $w = 4.0$. Class label = "bald eagle, American eagle, Haliaeetus leucocephalus" (22).

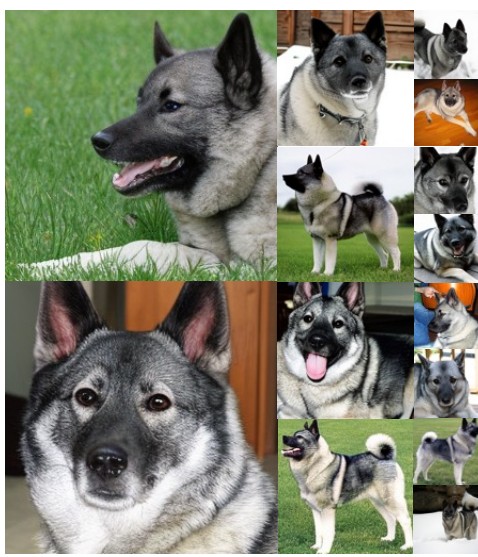

Figure 24: **Uncurated generation results of BFM-XL$_{\text{SF}}$**. We use classifier-free guidance with $w = 4.0$. Class label = "Norwegian elkhound, elkhound" (174).

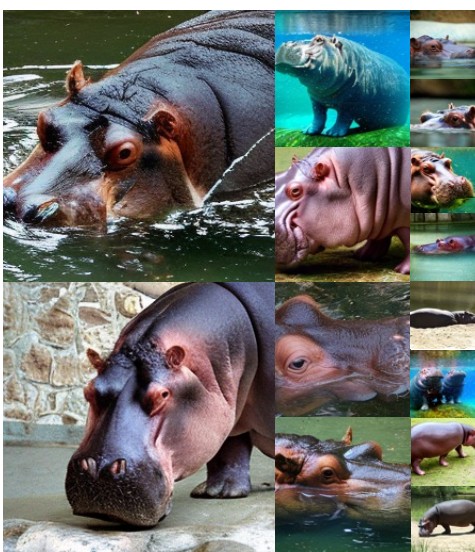

Figure 25: **Uncurated generation results of BFM-XL$_{\text{SF}}$**. We use classifier-free guidance with $w = 4.0$. Class label = "hippopotamus, hippo, river horse, Hippopotamus amphibious" (344).

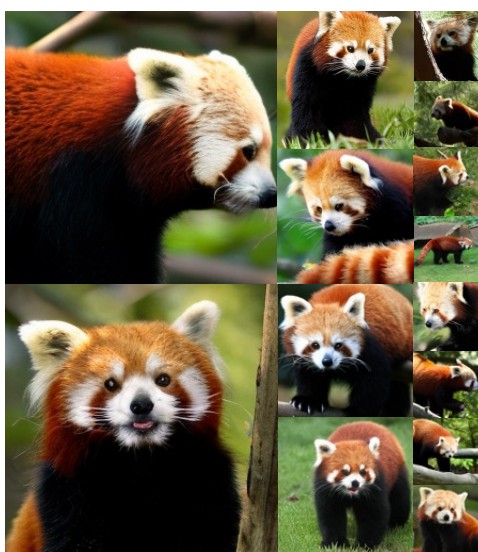

Figure 26: **Uncurated generation results of BFM-XL$_{\text{SF}}$**. We use classifier-free guidance with $w = 4.0$. Class label = "lesser panda, red panda, panda, bear cat, cat bear, Ailurus fulgens" (387).

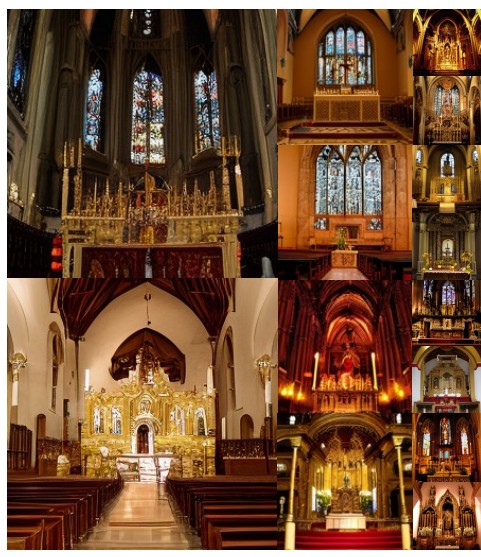

Figure 27: **Uncurated generation results of BFM-XL$_{\text{SF}}$**. We use classifier-free guidance with $w = 4.0$. Class label = "altar" (406).

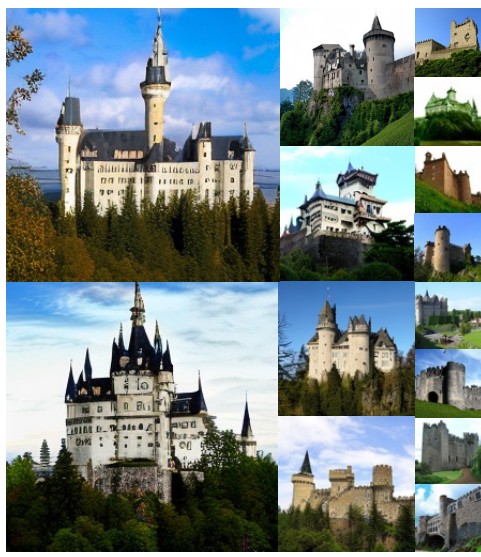

Figure 28: **Uncurated generation results of BFM-XL$_{\text{SF}}$**. We use classifier-free guidance with $w = 4.0$. Class label = "castle" (483).

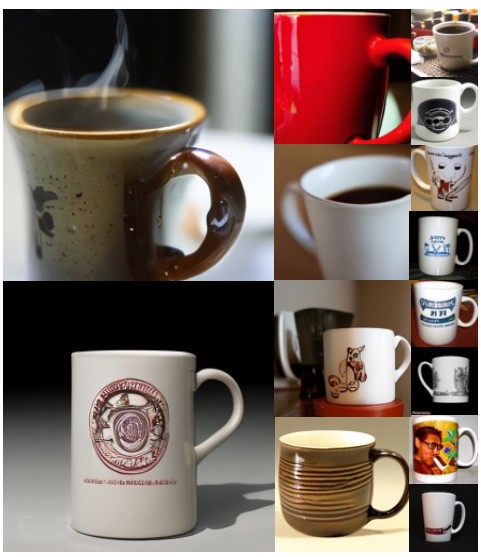

Figure 29: **Uncurated generation results of BFM-XL$_{\text{SF}}$**. We use classifier-free guidance with $w = 4.0$. Class label = "coffee mug" (504).

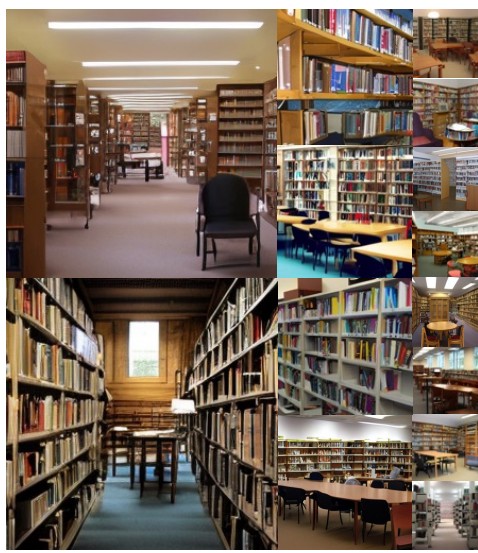

Figure 30: **Uncurated generation results of BFM-XL$_{\text{SF}}$**. We use classifier-free guidance with $w = 4.0$. Class label = "library" (624).

# E   Limitations and Future work

**Reliance on external features.** Our method relies on semantic features extracted from a pretrained visual encoder (e.g., DINOv2) to guide the velocity models through SemFeat. While this enables stronger semantic conditioning and improves generation quality, it introduces an external dependency and limits the model's generality to domains where such pretrained encoders are available. In future work, we plan to explore self-supervised alternatives that jointly learn semantic representations during training, potentially eliminating the need for external networks.

**Multi-modal large-scale datasets.** Our current evaluation focuses on single-modality image generation using ImageNet. While BFM scales well in this setting, applying the same framework to multi-modal, large-scale datasets (e.g., text-to-image or video datasets) remains an open challenge. Future research could investigate how to adapt blockwise modeling and semantic conditioning to settings that require more complex cross-modal reasoning and longer temporal consistency, such as text-driven video generation.

**Non-uniform partitioning schedule.** In this work, we adopt a uniform temporal segmentation strategy for simplicity. However, the generative trajectory exhibits varying complexity across timesteps, suggesting that certain regions (e.g., early noisy stages or late refinement phases) may benefit from finer-grained modeling. Future work could explore non-uniform or learnable partitioning schedules that allocate computational resources adaptively based on local signal complexity, potentially improving both performance and efficiency.

# F   Broader impact

The goal of this work is to improve the efficiency and scalability of generative models. By introducing Blockwise Flow Matching (BFM), we reduce the computational burden of high-quality generation, making diffusion-based models more accessible for researchers and practitioners with limited resources. This has potential societal benefits in democratizing access to generative AI, enabling smaller labs, startups, and educational institutions to experiment with cutting-edge models without requiring extensive computational infrastructure. However, improved efficiency also lowers the barrier for misuse. Generative models can be exploited to produce content such as deepfakes or misinformation at scale. We acknowledge that its capabilities could be repurposed in harmful ways. Mitigating such misuse requires responsible downstream deployment, as well as efforts from the broader community to develop and enforce ethical standards around generative AI.

