# OpenReview forum: "Blockwise Flow Matching: Improving Flow Matching Models For Efficient High-Quality Generation"
_NeurIPS.cc/2025/Conference — NeurIPS 2025 poster_

### Official Review · Reviewer_6WUg · 2025-06-17

**Clarity:** 3
**Significance:** 2
**Originality:** 2
**Rating:** 4
**Confidence:** 4

**Summary:**

This paper introduces Blockwise Flow Matching (BFM), a method that splits the generative trajectory in flow matching models into multiple temporal “blocks,” each with their own smaller, specialized network for a subset of timesteps. Combined with a Semantic Feature Guidance module (using pretrained DINOv2 features) and a lightweight Feature Residual Approximation for fast inference, BFM aims to improve both sample quality and efficiency. Experiments on ImageNet 256x256 show state-of-the-art FID at significantly reduced FLOPs, suggesting BFM dramatically pushes the quality/speed Pareto frontier.

**Questions:**

**Questions**

- In line 138, are the intervals uniform? Could the optimal non-uniform intervals in [2] provide any motivation or benefits to your design?

**References**

[1] Yan, Hanshu, et al. "Perflow: Piecewise rectified flow as universal plug-and-play accelerator." arXiv preprint arXiv:2405.07510 (2024).

[2] Nguyen, Bao, Binh Nguyen, and Viet Anh Nguyen. "Bellman optimal stepsize straightening of flow-matching models." arXiv preprint arXiv:2312.16414 (2023).

[3] Yoon, Jongmin, and Juho Lee. "Sequential Flow Straightening for Generative Modeling." arXiv preprint arXiv:2402.06461 (2024).

**Ethical Concerns:**

["NO or VERY MINOR ethics concerns only"]

**Final Justification:**

The author addresses my concern about the novelty of the paper. I decided to raise the score

**Limitations:**

yes

**Quality:**

3

**Strengths And Weaknesses:**

**Strengths**

- The paper is clear and easy to follow.
- Integrating pretrained semantic features makes sense and performs better than previous representation alignment techniques.
- Demonstrates robust improvements across quality and FLOPs, backed by systematic ablations.

**Weaknesses**

- I questioned the novelty of the BFM. Previous works have already divided the entire time interval into multiple segments and then optimized the inference quality for each of them. Some mentioned works can be listed as [1], [2], [3]. For example, the work [2] uses a similar loss function to yours; the difference is that they use only one network, which is specialized for all sub-segments. Similar principles can be found in [2] and [3]. I believed that deploying one neural network for inference is much resource-friendly than the paper’s approach. Could a distilled single model recover BFM’s benefits?

- The comparison with the works mentioned in the first drawback is highly recommended.
- Results are limited to ImageNet and strong pretrained image encoders; adaptation to other data types or domains isn’t addressed.
- Increased total parameters from block specialization may strain memory and training scalability, not fully discussed.
- The choice and allocation of temporal blocks lacks theory or automation—performance appears sensitive to segmentation hyperparameters.

---

> ### Author Rebuttal · Authors · 2025-07-29
>
> Thank you for the valuable feedback. We'd like to clarify the additional questions the reviewer raised.
>
> ---
> > ***Q1. The comparison with the relevant distillation-based methods [1,2,3] that aim to optimize the temporal steps.***
>
> We thank the reviewer for referring interesting works. We will include them in the final version with the discussion below.
>
> Our framework, Blockwise Flow Matching (BFM), is a **new architecture that can be trained from scratch.** In our experiment, we trained BFM from scratch. For more efficient and effective from-scratch training, we proposed a **new alignment method, SemFeat, tailored for BFM, using DINOv2**. To some extent, this is relevant to the prior works [1,2,3], but our framework does **NOT** leverage any pre-trained diffusion models.
> Indeed, **our approach is orthogonal and complementary to distillation-based methods.** Distillation aims to reduce the “number of sampling steps”, whereas BFM is focused on reducing the “computation cost per step” via architectural specialization. Importantly, distillation or schedule-optimization techniques can be incorporated into BFM to further accelerate inference if desired. Thus, BFM can serve as a strong foundation that benefits from further post-training improvements.
>
> To directly demonstrate BFM's *from-scratch* efficiency, we integrated our method into the recently published MeanFlow framework [4], which enables few-step sampling from scratch. Specifically, we adapted our method by creating four specialist blocks for the 4-step diffusion trajectory and applying the MeanFlow objective to each segment. After training, we can efficiently sample with only 4 steps (one per segment). The table below summarizes the results on ImageNet 256×256.
>
> | Model                   | NFE | FID  |
> |-------------------------|-----|------|
> | MeanFlow-B/2            | 4   | 13.2 |
> | Ours (MeanFlow-B/2+BFM) | 4   | 9.7  |
>
> This FID improvement confirms that BFM provides substantial gains as a from-scratch few-step methodology, independent of any distillation techniques.
>
> That said, we also recognize the importance of including relevant papers to provide more comprehensive understanding of the paper. We will cite the papers and include discussion in the revised manuscript.
>
> ---
>
> > ***Q2. Results are limited to ImageNet and strong pretrained image encoders; adaptation to other data types or domains isn’t addressed.***
>
> Thank you for this excellent point about the generality of our method. While our initial experiments focused on ImageNet for rigorous benchmarking, our framework is not limited to this setting. To demonstrate its versatility, we have conducted two significant new experiments.
>
> - **Scalability to Higher Resolution (ImageNet 512x512)**:
>
> To test scalability, we trained BFM on ImageNet 512×512 for 400K iterations under identical settings. As shown below, our method outperforms strong baselines, demonstrating robust performance at higher resolutions.
>
> | Method         | FID    | NFE |
> |----------------|--------|-----|
> | SiT-S/4        | 102.54 | 246 |
> | REPA-S/4       | 91.34 | 246 |
> | BFM-S/4 (Ours) | 87.85  | 246 |
>
> - **Generalizing to Text-to-Image Backbone (MS-COCO 256x256)**:
>
> To test generality, we integrated BFM into the state-of-the-art MMDiT [1] backbone for text-to-image synthesis. We trained on MS-COCO for 150K iterations (batch size 256), following the REPA setup. BFM again improves performance over both the original SOTA model and its REPA-augmented version.
> | Method         | FID  | NFE |
> |----------------|------|-----|
> | MMDiT          | 6.05 | 50  |
> | MMDiT + REPA   | 4.73 | 50  |
> | Ours (MMDiT + BFM)  | 4.32 | 48  |
>
> These two results confirm that BFM is a versatile technique that scales effectively to higher resolutions and generalizes to improve state-of-the-art, multi-modal architectures for text-to-image generation.
>
> ---
>
> > ***Q3. Increased total parameters from block specialization may strain memory and training scalability, not fully discussed.***
>
> Our Blockwise Flow Matching (BFM) can be viewed as sparse activation (a.k.a Mixture of Experts with a fixed router). Like MoEs, **our method increases parameters but maintains the computational cost/latency.** For comparison, we measured the training resource usage of BFM against strong baselines (SiT-XL/2, REPA-XL/2) on ImageNet 256×256, using 8 A100 GPUs with a batch size of 32 per GPU. As shown in the following table, **our model has the lowest GFLOPs and comparable GPU hours for training, while achieving the best FID score.** The quantitative results are as follows:
> | Model        | GFLOPs / Iteration | Train Hours (8*A100) | Memory (GB) | FID  | Params |
> |--------------|--------------------|--------------------|-------------|------|--------|
> | SiT-XL/2     | 22,789             | 53                 | 30          | 17.2 | 675M   |
> | REPA-XL/2    | 23,000             | 65                 | 31          | 7.9  | 675M   |
> | BFM-XL/2 (Ours)    | 21,816             | 57                 | 40          | 6.4  | 942M   |
> - **Training memory**: While BFM-XL/2 uses more parameters, the actual memory consumption is only moderately increased (from 30–31GB to 40GB), which remains practical for modern GPUs.
> - **Training efficiency**: The total training time is *comparable to or even less* than REPA-XL/2, and BFM achieves a substantially *better FID for the same number of iterations*.
> These results show that, despite increased total parameter count, BFM **remains practical and scalable** for training, and delivers **improvements in sample quality**. Also, each block is assigned to a pre-defined time interval; therefore, it is possible to train each block parallelly (and separately) across multiple GPUs, which can further improve the training scalability.
> We recognize this is a vital point for readers. We will add this table and a detailed discussion on the trade-offs between parameter count, training cost, and performance to the Appendix of our final paper.
> ---
>
> > ***Q4. The choice and allocation of temporal blocks lacks theory or automation—performance appears sensitive to segmentation hyperparameters.***
>
> This is an excellent point which is an interesting future direction, as discussed in "Limitation and Future work" section of our paper.
> Table 4 in our main paper shows that performance generally improves as the number of blocks increases. However, it leads to a larger model size. We empirically chose the number of temporal blocks based on the trade-off between performance and model size, but we agree that exploring a more theoretically-grounded or automated way to determine the optimal block segmentation is a promising direction for future research.
>
> ---
>
> > ***Q5. In line 138, are the intervals uniform? Could the optimal non-uniform intervals in [2] provide any motivation or benefits to your design?***
>
> Yes, in our current work, the time intervals are uniform. We chose this simple approach to provide a clean condition for our core contributions: the blockwise architecture and the semantic guidance module.
>
> We appreciate the reviewer's insightful suggestion. Using optimal non-uniform intervals, like those discussed in [2], is a fantastic idea and a natural next step for our framework. As we discussed in Limitation and Future work section, dynamically allocating more capacity (i.e., more or wider blocks) to more "difficult" parts of the generation trajectory could lead to significant further improvements in efficiency and quality.
>
> ---
> [1] Yan, Hanshu, et al. "Perflow: Piecewise rectified flow as universal plug-and-play accelerator." arXiv preprint arXiv:2405.07510 (2024). \
> [2] Nguyen, Bao, Binh Nguyen, and Viet Anh Nguyen. "Bellman optimal stepsize straightening of flow-matching models." arXiv preprint arXiv:2312.16414 (2023). \
> [3] Yoon, Jongmin, and Juho Lee. "Sequential Flow Straightening for Generative Modeling." arXiv preprint arXiv:2402.06461 (2024). \
> [4] Geng, Zhengyang, et al. "Mean flows for one-step generative modeling." arXiv preprint arXiv:2505.13447 (2025).

---

> > ### Comment · Reviewer_6WUg · 2025-08-05
> >
> > Thanks for your detailed responses. Your answers resolve almost all my concerns. I raise the score to reflect my more positive view. Two follow-up questions are:
> >
> > - Does the method increase the VRAM usage? I believed that the specialization for each step requires loading many specialized models to GPUs, leading to an increase in VRAM usage compared to using just one model conditioned on steps. Is using a memory-efficient component like LoRA on top of a base model for each specialized model better in this aspect?
> >
> > - After obtaining, for example, four specialized models for four steps, how can you adapt them if you want to use more steps, like 6, or fewer steps, like 2?

---

> ### Author Response · Authors · 2025-08-05
>
> Thank you for your positive feedback and for these thoughtful follow-up questions!
>
> ---
>
> > ***Does the method increase the VRAM usage? I believed that the specialization for each step requires loading many specialized models to GPUs, leading to an increase in VRAM usage compared to using just one model conditioned on steps. Is using a memory-efficient component like LoRA on top of a base model for each specialized model better in this aspect?***
>
> In our current implementation, all specialized blocks are loaded into GPU memory during both training and inference, resulting in higher VRAM usage compared to baseline models (as shown in the table above: 40GB for BFM-XL/2 versus 31GB for REPA-XL/2). However, because only the relevant block is active at each step, the computational cost (FLOPs) remains comparable to the baselines.
>
> We sincerely appreciate your suggestion of using parameter-efficient components like LoRA. Incorporating LoRA or similar adapter-based techniques on top of a shared base model for each block could significantly reduce VRAM and storage requirements, making the approach even more scalable. We see this as a promising direction for future work and thank you for the insightful suggestion.
>
> ---
>
> > ***After obtaining, for example, four specialized models for four steps, how can you adapt them if you want to use more steps, like 6, or fewer steps, like 2?***
>
> While we have not focused on this aspect yet, it is possible to re-allocate the existing specialized blocks to new segmentations based on the fractional overlap with the original intervals, and then finetune these blocks to recover optimal performance. Since each block has already been trained on a specific range of timesteps, adapting to a new segmentation generally requires less training than starting from scratch.
>
> ---
> Thank you once again for your constructive and invaluable feedback. Please let us know if you have any further concerns. We would be happy to clarify them!

---

### Official Review · Reviewer_wHnp · 2025-06-26

**Clarity:** 2
**Significance:** 3
**Originality:** 3
**Rating:** 5
**Confidence:** 3

**Summary:**

The authors of this work study a novel training scheme for flow matching models, which are state-of-the-art in image, text, etc, generation. In standard flow matching training, the practitioner has access to the start and end distributions and trains a neural network to learn vector fields at random times within the interval [0,1]. The hardest part in training is being able to capture the precise moments where the feature-learning takes place, and at inference, the need to discretize the flow into many steps is also expensive. To remedy these issues, this work proposes BlockwiseFlowMatching (BFM), where they train more (but smaller) specialized velocity blocks for particular time intervals. They compare their approach to several other works and they achieve SOTA results on large imagenet datasets.

**Questions:**

- Is it possible to distill the training ideas into a few key steps in the main text? I personally found Figure 2 and the accompanying caption very difficult to understand compared to the pseudocode in the appendix
- Do you attempt to see what happens when you take other paths?
- Following the "strengths" comment above, if they are available, is it possible to see what the images look like for various times (apart from just the semantic features)?
- From the semantic features: why does it look like the model very early on learns the image? This is a bit strange?
- Is Table1 inconsistent with the "lower is better"? I feel like it would be better if the first column said something like 0.66 and 1/1.09, and so on

**Ethical Concerns:**

["NO or VERY MINOR ethics concerns only"]

**Final Justification:**

All of my initial comments were addressed

**Quality:**

3

**Strengths And Weaknesses:**

A huge strength of this paper lies in its simplicity---the idea is very natural from an algorithmic/training perspective. My understanding is that many folks have tried to basically over- or under-tune training via stepsize schedulers, and I view this work as a complementary perspective on the matter (namely, to segment the time interval and train in block instead). I really appreciated the figures in the main text and appendix that highlight the "semantic features over timesteps"---this was very cool.

A weakness is that the work is purely empirical, but that is less of a weakness and more of just where the paper fits in the ML space. The clarity could be improved in some cases; see below. I am also not an expert, but it is not clear to me whether or not their experimentation is exhaustive enough with respect to the existing literature. Looking closer at Table 2, the number of parameters required is sometimes larger than other models (see e.g., the reference [45]) with comparable FID and the number of FLOPs is lower.

The numerical results are not entirely compelling and seem a bit catered for the datasets in question. That being said, I do appreciate the novelty this idea brings to this space, and I thus recommend weak accept.

---

> ### Author Rebuttal · Authors · 2025-07-30
>
> Thank you for the valuable feedback. We'd like to clarify the additional questions the reviewer raised.
>
> ---
>
> > ***Q1. It is not clear to me whether or not their experimentation is exhaustive enough with respect to the existing literature. The numerical results are not entirely compelling and seem a bit catered for the datasets.***
>
> We appreciate the reviewer for the concern about the scope of experiments. Our primary experiments were conducted on ImageNet to enable rigorous benchmarking and direct comparison with recent diffusion and flow models. To address the concern that our results are catered, we have conducted two significant new experiments on more demanding tasks beyond the initial ImageNet 256x256 benchmark.
>
> - **Scaling to Higher Resolution (ImageNet 512x512)**:
>
> To prove our method isn't just strong for moderate resolution, we tested it on higher-resolution 512x512 image generation. The results show a compelling and significant improvement over strong baselines.
>
> | Method         | FID    | NFE |
> |----------------|--------|-----|
> | SiT-S/4        | 102.54 | 246 |
> | REPA-S/4       | 91.34 | 246 |
> | BFM-S/4 (Ours) | 87.85  | 246 |
>
> Our method achieves over 3 FID improvement over the next best baseline, demonstrating strong performance and scalability.
>
> - **Generalizing to Text-to-Image Synthesis (MS-COCO)**:
>
> To prove our method's general applicability beyond class-conditional generation, we integrated it into a state-of-the-art text-to-image backbone (MMDiT [1]) and tested it on the MS-COCO dataset.
>
> | Method         | FID  | NFE |
> |----------------|------|-----|
> | MMDiT          | 6.05 | 50  |
> | MMDiT + REPA   | 4.73 | 50  |
> | Ours (MMDiT + BFM)  | 4.32 | 48  |
>
> These results confirm that BFM scales robustly to higher resolutions and adapts seamlessly to modern, multi-modal transformer architectures. Although we have not yet conducted experiments at the largest model scales due to hardware limits, these results demonstrate that our framework is fundamentally modular and can be integrated with larger, state-of-the-art backbones.
>
> ---
>
> > ***Q2. Is it possible to distill the training ideas into a few key steps in the main text? I personally found Figure 2 and the accompanying caption very difficult to understand compared to the pseudocode in the appendix.***
>
> Thank you for your helpful feedback regarding the clarity of Figure 2 and the training procedure. In the revised version, we will summarize the training pipeline in the main text as a concise list of key steps and revise Figure 2 and its caption for greater clarity. We will also directly reference the pseudocode in the appendix for readers seeking more detailed implementation guidance.
>
> ---
>
> > ***Q3. Do you attempt to see what happens when you take other paths?***
>
> Thank you for this excellent question. We interpret "other paths" as referring to different probability trajectories between noise and data (e.g., non-linear paths vs. our default linear one). To test this, we applied our BFM framework to a DiT model that uses the standard DDPM path. The results clearly show that our method provides a significant benefit even when applied to this different, non-linear trajectory. The results are summarized below:
>
> | Method              | FID  |
> |---------------------|------|
> | DiT-S/2 (DDPM)      | 90.4 |
> | BFM-S/2 (DDPM)      | 77.2 |
>
> This 13.2-point FID improvement demonstrates that the benefits of our method are robust and not limited to a specific type of probability path.
>
> ---
>
> > ***Q4.  If they are available, is it possible to see what the images look like for various times (apart from just the semantic features)?***
>
> We thank the reviewer for this excellent suggestion. Visualizing the generated images at various timesteps along the diffusion trajectory can provide valuable insight into how BFM operates and how different blocks contribute to the final sample. In the revision, we will include visualizations of intermediate generations at multiple timesteps.
>
> ---
> > ***Q5. From the semantic features: why does it look like the model very early on learns the image? This is a bit strange?***
>
> This is an insightful observation that highlights a key feature of our semantic feature. Our alignment network learns to predict the final image's semantic features very early on, even from highly noisy inputs. This is the intended behavior and a core strength of our design.
>
> Our alignment network is explicitly trained to match the features of the clean image regardless of the input timestep, which learns to be a powerful semantic denoiser. As a result, the network learns to robustly extract semantic information from noisy images. This provides a strong, consistent semantic feature that guides all the specialized velocity blocks throughout the entire generation process, which we believe is a primary reason for our method's strong performance.
>
> We acknowledge this is an interesting phenomenon and will discuss it in the revised manuscript.
>
> ---
>
> > ***Q6. Is Table1 inconsistent with the "lower is better"? I feel like it would be better if the first column said something like 0.66 and 1/1.09, and so on.***
>
> Thank you for your suggestion regarding the presentation of Table 1. We structured Table 1 to follow the logical flow of our method section—first introducing blockwise flow matching, then semantic feature guidance, and finally the feature approximation network—so that the progression of methods aligns with the corresponding impact on fidelity and efficiency.
>
> However, we appreciate your feedback on clarity. In the revised version, we will consider reformatting Table 1 to make the “lower is better” interpretation more transparent (for example, by using relative ratios such as 0.66 or 1/1.09) as you suggested. Thank you for helping us improve the presentation of our results.
>
> ---
>
> [1] Esser, Patrick, et al. "Scaling rectified flow transformers for high-resolution image synthesis." Forty-first international conference on machine learning. 2024 \

---

> > ### Comment · Reviewer_wHnp · 2025-08-04
> >
> > Thank you for addressing all of my comments! I have decided to raise my score

---

### Official Review · Reviewer_5kd7 · 2025-06-30

**Clarity:** 3
**Significance:** 3
**Originality:** 3
**Rating:** 4
**Confidence:** 5

**Summary:**

The paper proposes partitioning the original diffusion process into multiple time intervals, where each interval is managed by a separate, compact network. This design reduces the model feedforward cost incurred at every evaluation step.

**Questions:**

Q1. Why FRN is needed? Why is not not trained jointly with feature alignment network?

Q2. Using non-overlapping time interval is reasonable since it is easier to manage. However, it also has a limitation that there is a disconnected information at the starting and ending point between each interval. Have the authors considered this issue in the method?

Q3. In semantic feature guidance, what is the model performance when only aligment loss is used and no conditioning semantic feature $f_t$ is used. This allows to compare with the naive REPA baseline.

Q4. As concern in Q2, this seems that the authors addressed a different issue via Feature Residual Approximation which is the consistency of starting point and intermediate point within the same interval. However, there is an inevitable discrepancy between the ending point of the previous time interval and the starting point of the next time interval, even though they should theoretically be identical. The feature residual only addresses the discrepancy of within-interval features, not the temporal discontinuity between consecutive intervals.

Q5. Have the authors tried a simple baseline that aligns the features of velocity block with pretrained features, without the presence of Semantic module? This is similar to REPA method.

Q6. Feature residual approximation is just one approach. Have the authors tried a simple interpolation between the start and end points instead so there is no need of additional learnt module.

**Ethical Concerns:**

["NO or VERY MINOR ethics concerns only"]

**Final Justification:**

The rebuttal and follow-up response from the authors generally addressed my major concerns about the discontinuity between adjacent segments. So, I raised my score to reflect this. The paper's contribution is substantial with strong performance advantage (both speed and quality) which could facilitate for the next AI generation models.

**Limitations:**

yes but please also address the concern of discontinuity of non-overalaping inverals above.

**Quality:**

3

**Strengths And Weaknesses:**

Strenths:
- Decomposing a cumbersome network into multiple smaller ones is a sensible idea where each handles an assigned temporal block.
- This gives a flexibility in eliminating computational cost while obtaining good generation quality.
- The exposition is clear and well-motivated.

Weaknesses:
- The proposed components, including semantic guidance and the feature residual module, add considerable complexity to the method and could benefit from clearer motivation to justify their inclusion.
- There is a compromise of model performance when using the Feature Residual Approximation which is understandable. I would be helpful to include a speed vs. FID chart for demonstration of the tradeup.

---

> ### Author Rebuttal · Authors · 2025-07-31
>
> Thank you for the valuable feedback. We'd like to clarify the additional questions the reviewer raised.
>
> ---
>
> > ***Q1. The proposed components, including semantic guidance and the feature residual module, add considerable complexity to the method and could benefit from clearer motivation to justify their inclusion. Why FRN is not not trained jointly with feature alignment network?***
>
> We thank the reviewer for asking for a clearer justification of our design choices.
> We have provided a detailed motivation in Sections 4.2–4.3 and Table 1 in the main paper, but we are happy to clarify here:
>
> Our two main components are designed to solve specific, practical challenges in the blockwise modeling framework.
> - **Semantic Feature Guidance**: The velocity blocks operating at early timesteps primarily encounter highly noisy samples, limiting their ability to learn high-level semantic structure from clean data. Our solution is a shared alignment network that provides a consistent, global semantic understanding to all blocks. This encourages *smooth transitions and rich semantic guidance* throughout the diffusion process.
> Moreover, this module adds minimal complexity by following an encoder–decoder structure widely used in modern architectures.
>
> - **Feature Residual Network**: The problem is that repeatedly running the large guidance network during inference is inefficient. Our solution is the FRN, a very lightweight network that learns to approximate the guidance features locally within each block. This eliminates the need to call the alignment network at every inference step. FRN is a lightweight, optional module: BFM with semantic feature guidance alone is already powerful for high-fidelity generation, and FRN simply provides an efficient pathway for inference when computational resources are constrained.
>
> - **Why not train jointly?** If trained together, the shared alignment network might learn features that are easier for the FRN to approximate, rather than learning the best possible features for guiding generation. By using a two-stage process, we first train the guidance network to be as powerful as possible without constraints. Then, in a second stage, we train the lightweight FRN to mimic this optimal. We found empirically that this staged approach leads to more stable training and higher final sample quality.
>
> In summary, both the semantic feature guidance module and FRN address concrete challenges in blockwise flow modeling, ensuring global semantic coherence and practical inference efficiency.
>
> ---
>
> > ***Q2. Using non-overlapping time interval is reasonable since it is easier to manage. However, it also has a limitation that there is a disconnected information at the starting and ending point between each interval. Have the authors considered this issue in the method?***
>
> We thank the reviewer for this insightful question about potential discontinuities at our block boundaries.
> Our design includes two mechanisms that inherently mitigate this issue:
> - Our framework is designed so that the marginal distributions at the endpoint of one segment and the starting point of the next segment are matched exactly. This theoretically prevents any explicit discontinuity in the underlying probability trajectory.
> - Furthermore, our alignment network is shared across all segments and learns features over the entire diffusion process, which encourages smooth transitions and consistency between blocks.
>
> Empirically, we have not observed any visual artifacts or performance degradation that could be attributed to boundary discontinuities, which we believe is due to these two design choices.
>
> We acknowledge the possibility of subtle transitions at segment boundaries. In future work, we plan to explore mechanisms to further encourage temporal continuity between blocks, such as overlapping temporal segments.
>
> ---
>
> > ***Q3. In semantic feature guidance, what is the model performance when only aligment loss is used and no conditioning semantic feature is used. This allows to compare with the naive REPA baseline.***
>
> We thank the reviewer for this excellent suggestion, which helps to isolate the specific contribution of our blockwise architecture. As requested, we ran an experiment using BFM but replaced our explicit guidance module with REPA-style alignment loss, using DINOv2 as the target. This provides a direct comparison to a standard Flow Matching (FM) model with REPA.
>
> |               | FID   |
> |---------------|-------|
> | FM + REPA      | 72.9  |
> | BFM + REPA      | 70.23 |
>
> The results show that even without our proposed semantic conditioning, the blockwise architecture itself provides a clear performance improvement over the standard baseline. This demonstrates the inherent benefit of using specialized temporal blocks. We will include and clarify these results in the revised manuscript.
>
> ---
>
> > ***Q4. As concern in Q2, this seems that the authors addressed a different issue via Feature Residual Approximation which is the consistency of starting point and intermediate point within the same interval. However, there is an inevitable discrepancy between the ending point of the previous time interval and the starting point of the next time interval, even though they should theoretically be identical. The feature residual only addresses the discrepancy of within-interval features, not the temporal discontinuity between consecutive intervals.***
>
> We thank the reviewer for this follow-up question. We would like to clarify that the Feature Residual Network (FRN) is *not* introduced to address the consistency between the starting and intermediate points within the same interval. Instead, its primary purpose is to enable efficient inference by approximating the output of the alignment network, thus reducing computational overhead at test time.
>
> The issue of continuity between blocks is handled by two other core design choices, as mentioned in Q2: 1)
> shared alignment network, and 2) identical marginal distribution at boundaries.
>
> ---
>
> > ***Q5. Have the authors tried a simple baseline that aligns the features of velocity block with pretrained features, without the presence of Semantic module? This is similar to REPA method.***
>
> Thank you for your question. This is a crucial ablation that we also addressed in our response to Q3 above. As described in our previous answer, we have ablated the semantic module and evaluated the performance of BFM when using only the REPA-style alignment loss, directly aligning velocity block features with pretrained features.
>
> ---
>
> > ***Q6. Feature residual approximation is just one approach. Have the authors tried a simple interpolation between the start and end points instead so there is no need of additional learnt module.***
>
> Thank you the insightful suggestion. While interpolating between the start and end points of each interval is an appealing and efficient idea, it is unfortunately not feasible during inference: the end point of the interval corresponds to a future state, which is precisely what the model is tasked with generating. As a result, we cannot access this information during sampling.
>
> This is precisely why our Feature Residual Network (FRN) is necessary. Instead of interpolating between two known points, the FRN acts as a feature predictor, conditioned only on the information available up to the current step. This enables efficient and accurate inference without requiring knowledge of future states.

---

> > ### Comment · Reviewer_5kd7 · 2025-08-04
> >
> > Thank authors for detailed response! However, some concerns remain:
> >
> > > Q2. Our framework is designed so that the marginal distributions at the endpoint of one segment and the starting point of the next segment are matched exactly. This theoretically prevents any explicit discontinuity in the underlying probability trajectory.
> >
> > Could authors elaborate on this? since it could not see how the proposed method addresses the disconuity.
> >
> > > Q3. What is number of training iterations and number of generated samples for computing FID?
> >
> > > Q4. Since the FRA module approximates any intermediate output feature based on the feature at the starting point of one segment, I believe it still has the implicit effect of forcing intermediate features to align with features at the starting point. Although its original goal was to approximate output from a feature alignment network.

---

> ### Author Response · Authors · 2025-08-05
>
> We truly appreciate your thoughtful follow-up and request for clarification.
>
> ---
>
> > ***Could authors elaborate on this? since it could not see how the proposed method addresses the disconuity.***
>
> At training time, the endpoint of one segment and the starting point of the next segment are, by construction (see Eq. (4) and (5) in the main paper), both sampled from the same forward process and therefore share the same marginal distribution. This setup ensures that each model learns to match the marginal distribution at each boundary point, providing statistical continuity across segments. Therefore, it is statistically guaranteed that the generated output at the end of a segment will be consistent, *in distribution*, with the expected input of the next segment.
>
> This property is fully consistent with standard diffusion/flow-matching modeling, where each step is trained to match the correct marginal distribution, and explicit sample-wise continuity is not enforced. In both standard and our BFM models, the continuity between steps or segments is guaranteed in distribution, which has been empirically sufficient for smooth, high-quality generation.
>
> ---
>
> > ***What is number of training iterations and number of generated samples for computing FID?***
>
> Thank you for your question regarding experimental details. Unless stated otherwise, all models were trained for 100,000 iterations. For FID evaluation, we followed standard protocols and generated 50,000 samples for each model. We will make sure these details are clearly stated in the revised manuscript.
>
> ---
>
> > ***Since the FRA module approximates any intermediate output feature based on the feature at the starting point of one segment, I believe it still has the implicit effect of forcing intermediate features to align with features at the starting point. Although its original goal was to approximate output from a feature alignment network.***
>
> Thank you for raising this point about the Feature Residual Approximation (FRA) module and its interaction with the feature alignment network.
> It is true that the output of the FRA may sometimes resemble the starting point, since the FRA takes the starting feature as input.
>
> However, the alignment network is frozen during FRA training, so there is no feedback that would implicitly force or regularize its outputs to become closer to the starting point feature.
> As a result, FRA learns to best approximate the feature trajectory that the alignment network has already learned, without influencing the alignment network itself.
>
> ---
>
> Thank you once again for your constructive and invaluable feedback. Please let us know if you have any further concerns. We would be happy to clarify them!

---

> > ### Comment · Reviewer_5kd7 · 2025-08-05
> >
> > Thank you for clarifying those points! I'm satisfied with the response so I increased the score accordingly.

---

### Official Review · Reviewer_ByUL · 2025-07-02

**Clarity:** 3
**Significance:** 3
**Originality:** 3
**Rating:** 4
**Confidence:** 4

**Summary:**

This paper introduces Blockwise Flow Matching (BFM), a novel framework to address the efficiency and performance limitations of Flow Matching models. It partitions the generative trajectory into segments, each handled by a smaller, specialized network block to improve both modeling accuracy and inference speed. Furthermore, by incorporating semantic guidance and an efficient feature approximation strategy, the method achieves both quality and speed improvement. Overall, this paper proposes a brand new paradigm for Flow Matching models.

**Questions:**

1. In Equation 7, the authors supervise the denoising process by velocity within the $m$-th segment. I wonder whether supervising by velocity to $x_1$, namely, $\frac{x_{t_m}-x_1}{t_m-1}$ will result in performance degradation.
2. It is suggested to provide an ablation study where the feature alignment network is trained without the $\mathcal{L}_{align}$, which demonstrates the benefits of explicitly aligning to pretrained representations.
3. In Section 4.2 (Line 163), the authors mention that BFM can take various additional conditions as inputs (e.g., class labels or text prompts). However, I only see class-label-conditioned generation in the paper. Can the authors provide analysis regarding text-conditioned generation?
4. The design of the Semantic Feature Guidance module is similar to recent work DDT [1]. The authors are advised to add a citation and discussion.
5. It would be better if the authors provided a discussion or visualization of typical failure cases to demonstrate the limitations of BFM.

[1] Wang, Shuai, et al. "Ddt: Decoupled diffusion transformer." arXiv preprint arXiv:2504.05741 (2025).

**Ethical Concerns:**

["NO or VERY MINOR ethics concerns only"]

**Final Justification:**

I thank the author for patiently addressing my questions, and I have decided to improve my score.

**Limitations:**

yes

**Paper Formatting Concerns:**

Not found.

**Quality:**

3

**Strengths And Weaknesses:**

Strengths
1. The main idea of this paper, partitioning the generative trajectory into segments handled by specialized, smaller blocks, is a novel design and effectively deals with the temporal heterogeneity of the generation process.
2. The experiments are rigorous, comprehensive, and conducted on a competitive benchmark (ImageNet 256x256). The detailed ablation studies and visualizations systematically validate the contribution of each component of the proposed method, demonstrating high-quality research practice.
3. This is a well-organized paper. The authors do an excellent job of motivating the problem, explaining their method, and presenting their results in an accessible manner.

Weaknesses
1. Some key experiments validating the effectiveness of the proposed modules are missing, which I elaborate on in my Questions.

---

> ### Author Rebuttal · Authors · 2025-07-29
>
> Thank you for the valuable feedback. We'd like to clarify the additional questions the reviewer raised.
>
> ---
>
> > ***Q1. In Equation 7, the authors supervise the denoising process by velocity within the m-th segment. I wonder whether supervising by velocity to $x_1$, namely, $({x_{t_m} - x_1})/({t_m - 1})$ will result in performance degradation.***
>
> We thank the reviewer for this insightful question about the formulation of our objective. The two supervision targets are, in fact, mathematically identical. From the equations (4) and (5) in our paper, the endpoints $x_{t_{m-1}}$ and $x_{t_m}$ are defined as linear interpolations between $x_0$ and $x_1$:
>
> $$
> x_{t_{m-1}} = (1-t_{m-1})x_0 + t_{m-1}x_1, \quad x_{t_m} = (1-t_{m})x_0 + t_{m}x_1.
> $$
>
> Therefore,
> $$
> \frac{x_{t_m} - x_{t_{m-1}}}{t_m - t_{m-1}} = \frac{[(1-t_{m})x_0 + t_{m}x_1] - [(1-t_{m-1})x_0 + t_{m-1}x_1]}{t_m - t_{m-1}}
> $$
> $$
> = \frac{(t_{m-1}-t_m)x_0 + (t_m - t_{m-1})x_1}{t_m - t_{m-1}} = -x_0 + x_1 = x_1 - x_0.
> $$
>
> Similarly,
> $$
> \frac{x_{t_m} - x_1}{t_m - 1} = \frac{[(1-t_{m})x_0 + t_{m}x_1] - x_1}{t_m - 1} = \frac{(1-t_m)x_0 + (t_m-1)x_1}{t_m - 1} = x_1 - x_0.
> $$
>
> Therefore, there is no performance difference because the supervision signal is identical in both cases. We chose our formulation for notational consistency within the block-wise framework.
>
> ---
>
> > ***Q2. It is suggested to provide an ablation study where the feature alignment network is trained without the alignment loss, which demonstrates the benefits of explicitly aligning to pretrained representations.***
>
> We thank the reviewer for this excellent suggestion to empirically validate our design. As requested, we conducted an ablation study where the feature alignment network was trained without the alignment loss, on ImageNet 256x256 for 100K iterations:
>
> | Method     | Alignment loss | FID  |
> |------------|---------------|------|
> | BFM-S/2    | x             | 73.5 |
> | BFM-S/2    | o             | 66.9 |
>
> As shown, removing the alignment loss results in a substantial drop in sample quality (6.6 FID points), clearly demonstrating the effectiveness of explicit alignment to pretrained representations.
>
> We will add this ablation study to the Experiments section of our revised manuscript to quantitatively demonstrate the importance of the alignment loss.
>
> ---
>
> > ***Q3. Can the authors provide analysis regarding text-conditioned generation?***
>
> We thank the reviewer for this important question. To demonstrate our method's effectiveness on text-conditioned generation, we integrated BFM into the state-of-the-art MMDiT [1] backbone and trained it on the MS-COCO benchmark for 150K iterations (batch size 256), following the REPA setup. While testing on billion-scale models remains future work, this experiment shows BFM's compatibility with and ability to improve a modern, multi-modal architecture.
>
> | Method         | FID  | NFE |
> |----------------|------|-----|
> | MMDiT          | 6.05 | 50  |
> | MMDiT + REPA   | 4.73 | 50  |
> | Ours (MMDiT + BFM)  | 4.32 | 48  |
>
> As the results show, our BFM-augmented model surpasses both the original MMDiT and the REPA-enhanced version, achieving a stronger FID score. These results confirm that BFM is robust and adaptable to modern transformer-based multi-modal architectures.
>
> ---
>
> > ***Q4. The design of the Semantic Feature Guidance module is similar to recent work DDT [2]. The authors are advised to add a citation and discussion.***
>
> We thank the reviewer for pointing out this highly relevant concurrent work. We agree that DDT shares the high-level principle of using an explicit alignment module to guide generation.
> Our key distinction lies in the architectural integration and purpose. While DDT applies guidance to a standard diffusion model, our Semantic Feature Guidance is a core component of our modular, blockwise flow matching framework. It specifically provides high-level conditioning to a set of specialized, time-step-dependent velocity networks.
>
> We will update our Related Work section to include a citation and a detailed discussion of DDT. We will clarify the conceptual similarities while also highlighting the unique architectural differences of our approach.
>
> ---
>
> > ***Q5. It would be better if the authors provided a discussion or visualization of typical failure cases to demonstrate the limitations of BFM.***
>
> Thank you for this valuable suggestion. While our method significantly improves on baselines, it can still produce failure cases (e.g., minor artifacts or implausible object structures) that are common to most generative models trained on complex datasets like ImageNet. We have not observed failure modes that are uniquely attributable to our blockwise design or semantic guidance.
>
> We will include a curated set of generated images that exhibit typical failure modes and discuss them in the context of general challenges in generative modeling.
>
> ---
> [1] Esser, Patrick, et al. "Scaling rectified flow transformers for high-resolution image synthesis." Forty-first international conference on machine learning. 2024 \
> [2] Wang, Shuai, et al. "Ddt: Decoupled diffusion transformer." arXiv preprint arXiv:2504.05741 (2025).

---

> > ### Comment · Reviewer_ByUL · 2025-08-05
> >
> > I thank the author for patiently addressing my questions, and I have decided to improve my score.

---

### Official Review · Reviewer_ySpd · 2025-07-06

**Clarity:** 3
**Significance:** 3
**Originality:** 2
**Rating:** 4
**Confidence:** 4

**Summary:**

The paper presents an interesting approach to flow-matching based methods by partitioning the generative trajectory into multiple temporal segments. Each segment is handled by a specialized smaller model for prediction. The proposed algorithm is called blockwise flow matching. Experimental results on ImageNet shows reasonable accelerations, like up to 4.9x over the baseline, without compromising the accuracy.

**Questions:**

The main questions are in the experimental section. Please well address the comments in the weakness section.

**Ethical Concerns:**

["NO or VERY MINOR ethics concerns only"]

**Final Justification:**

Thanks for the authors' detailed rebuttal. Most of my concerns are cleared. I recommend borderline accept as the final rating.

**Limitations:**

The paper does not explicitly provide the limitations of the proposed algorithm. Please reference the weakness section above for more details listed.

**Paper Formatting Concerns:**

No.

**Quality:**

3

**Strengths And Weaknesses:**

### Strengths
- The idea of using blockwise flow matching is well motivated. As the paper claims, there are different characteristics for different timesteps during the flow trajectory. For example, early timesteps are dominated by irregular, low frequency patterns while the later timesteps require high-frequency details. Thus, using specialized models for different timesteps is reasonable.

- The design of the proposed algorithm in Fig. 2 can well reduce the computation cost of the vanilla flow-matching based algorithm while maintaining feasible performance during the generation process.

- The presentation of the paper is clear. The paper provides sufficient implementation details which are easy to reproduce the results.

- The paper presents extensive ablation studies to validate the effectiveness of the proposed algorithm.


### Weaknesses

- Although the proposed algorithm reports reasonable speed up (based on FLOPs) as well as superior performance gain over the baseline, as shown in Table 2, blockwise flow matching actually increases the model parameters. Thus it will also consume more training memory, which makes the training cost larger than the vanilla flow matching algorithms. Besides, for the inference time, the decrease of FLOPs does not mean the model will have similar speed-up gain. Is it possible to provide the inference time cost for these algorithms reported in Table 2？

- The proposed algorithm is based on ImageNet 256x256. How about the performance on the challenging text-to-image benchmark with larger generation size, like 1024x1024. Also, currently, the algorithm is based on the small model like SiT and REPA. Is it possible to be used in state-of-art image generation models?

- For the performance evaluation, FID may not be consistent with human preference. Is it possible to provide a user-study to further validate the effectiveness of the proposed algorithm.

- Is the proposed algorithm applicable to the algorithms with smaller diffusion steps, like 4-step model?

---

> ### Author Rebuttal · Authors · 2025-07-29
>
> Thank you for the valuable feedback. We'd like to clarify the additional questions the reviewer raised.
>
> ---
> > ***Q1. Blockwise flow matching actually increases the model parameters. Thus, it will also consume more training memory, which makes the training cost larger than the vanilla flow matching algorithms.***
>
> Our Blockwise Flow Matching (BFM) can be viewed as sparse activation (a.k.a Mixture of Experts with a fixed router). Like MoEs, **our method increases parameters but maintains the computational cost/latency.** For comparison, we measured the training resource usage of BFM against strong baselines (SiT-XL/2, REPA-XL/2) on ImageNet 256×256, using 8 A100 GPUs with a batch size of 32 per GPU. As shown in the following table, **our model has the lowest GFLOPs and comparable GPU hours for training, while achieving the best FID score.** The quantitative results are as follows:
> | Model        | GFLOPs / Iteration | Train Hours (8*A100) | Memory (GB) | FID  | Params |
> |--------------|--------------------|--------------------|-------------|------|--------|
> | SiT-XL/2     | 22,789             | 53                 | 30          | 17.2 | 675M   |
> | REPA-XL/2    | 23,000             | 65                 | 31          | 7.9  | 675M   |
> | BFM-XL/2 (Ours)    | 21,816             | 57                 | 40          | 6.4  | 942M   |
> - **Training memory**: While BFM-XL/2 uses more parameters, the actual memory consumption is only moderately increased (from 30–31GB to 40GB), which remains practical for modern GPUs.
> - **Training efficiency**: The total training time is *comparable to or even less* than REPA-XL/2, and BFM achieves a substantially *better FID for the same number of iterations*.
> These results show that, despite increased total parameter count, BFM **remains practical and scalable** for training, and delivers **improvements in sample quality**. Also, each block is assigned to a pre-defined time interval; therefore, it is possible to train each block parallelly (and separately) across multiple GPUs, which can further improve the training scalability.
> We recognize this is a vital point for readers. We will add this table and a detailed discussion on the trade-offs between parameter count, training cost, and performance to the Appendix of our final paper.
>
> ---
> > ***Q2. For the inference time, the decrease of FLOPs does not mean the model will have similar speed-up gain. Is it possible to provide the inference time cost?***
>
> We thank the reviewer for this excellent point. We agree that theoretical FLOPs do not always translate to real-world speed-ups. To provide a concrete answer, we have benchmarked the wall-clock inference time of our models against the baselines on an A100 GPU with a batch size of 32. The results are summarized below:
>
> | Method                | NFE | wallclock run-time (s) | FLOPs (G) | FID  |
> |-----------------------|-----|-----------------------|-----------|------|
> | SiT-XL/2              | 246 | 51.21                 | 114.5     | 2.08 |
> | REPA-XL/2             | 246 | 51.21                 | 114.5     | 1.46 |
> | BFM-XL/2-SF (Ours)    | 246 | 47.96                 | 107.8     | 1.36 |
> | BFM-XL/2-SF-RA (Ours) | 246 | 19.42                 | 37.8      | 2.03 |
>
> These empirical results confirm our method's efficiency:
> - Our primary model, BFM-SF is **7%** faster than the baselines while also achieving a superior FID score.
> - Our most efficient variant, BFM-SF-RA, achieves **63%** reduction in runtime while maintaining an FID score comparable to the SiT baseline.
>
> This confirm that the observed drop in FLOPs with BFM is directly reflected in real-world speed-up, validating the practical efficiency of our approach. We will add this wall-clock time comparison to the main experimental section of our paper to demonstrate the practical efficiency of our method.
>
> ---
> > ***Q3. How about the performance on the challenging text-to-image benchmark with larger generation size, like 1024x1024. Also, currently, the algorithm is based on the small model like SiT and REPA. Is it possible to be used in state-of-art image generation models?***
>
> We thank the reviewer for these important questions on the scalability and generality of our method. While we have not yet run experiments on the billion-parameter models due to current resource constraints, our framework is fundamentally modular and can be integrated with larger, state-of-the-art backbones. To demonstrate scalability, we conducted two new experiments to address the reviewer's specific points:
>
> - **Scalability to Higher Resolution (ImageNet 512x512)**:
>
> To test scalability, we trained BFM on ImageNet 512×512 for 400K iterations under identical settings. As shown below, our method outperforms strong baselines, demonstrating robust performance at higher resolutions.
>
> | Method         | FID    | NFE |
> |----------------|--------|-----|
> | SiT-S/4        | 102.54 | 246 |
> | REPA-S/4       | 91.34 | 246 |
> | BFM-S/4 (Ours) | 87.85  | 246 |
>
> - **Generalizing to Text-to-Image Backbone (MS-COCO 256x256)**:
>
> To test generality, we integrated BFM into the state-of-the-art MMDiT [1] backbone for text-to-image synthesis. We trained on MS-COCO for 150K iterations (batch size 256), following the REPA setup. BFM again improves performance over both the original model and its REPA-augmented version.
> | Method         | FID  | NFE |
> |----------------|------|-----|
> | MMDiT          | 6.05 | 50  |
> | MMDiT + REPA   | 4.73 | 50  |
> | Ours (MMDiT + BFM)  | 4.32 | 48  |
>
> These two results confirm that BFM is a versatile technique that scales effectively to higher resolutions and generalizes to improve state-of-the-art, multi-modal architectures for text-to-image generation.
>
> ---
>
> > ***Q4. For the performance evaluation, FID may not be consistent with human preference. Is it possible to provide a user-study to further validate the effectiveness of the proposed algorithm.***
>
> We fully agree that while FID is a widely used quantitative metric, it does not always align perfectly with human judgments of realism and visual preference. To provide a more comprehensive assessment, we have demonstrated additional quantitative metrics, *Inception Score, S-FID, Precision, and Recall* in the main paper, which measure different aspects of generation quality.
>
> In response to your suggestion, we will consider including a user study where human evaluators perform blind, pairwise comparisons between our method's generations and those from the strongest baseline.
>
> > ***Q5. Is the proposed algorithm applicable to the algorithms with smaller diffusion steps, like 4-step model?***
>
> Thank you for this insightful question that probes the generality of our method in the challenging few-step sampling regime.
> To validate this, we integrated our method to the recently published MeanFlow [3] framework, an approach designed for training few-step diffusion model from scratch.
>
> Specifically, we adapted our method by creating four specialist blocks for the 4-step diffusion trajectory and applying the MeanFlow objective to each. After training, we enable efficient sampling with only 4 steps (one per segment). The table below summarizes the results on ImageNet 256×256:
>
> |        Model        | NFE | FID  |
> |----------------|-----|------|
> | MeanFlow-B/2   | 4   | 13.2 |
> | Ours (MeanFlow-B/2+BFM)	 | 4   | 9.7 |
>
>  As shown, BFM achieves significantly better FID. This experiment confirms that BFM is not only compatible with few-step frameworks but provides a significant performance advantage.
>
> Given the strength of this result, we will add this experiment in the revised manuscript to highlight BFM's applicability to the few-step generation setting.
>
> ---
> [1] Esser, Patrick, et al. "Scaling rectified flow transformers for high-resolution image synthesis." Forty-first international conference on machine learning. 2024 \
> [2] Radford, Alec, et al. "Learning transferable visual models from natural language supervision." International conference on machine learning. PmLR, 2021 \
> [3] Geng, Zhengyang, et al. "Mean flows for one-step generative modeling." arXiv preprint arXiv:2505.13447 (2025)

---

### Note · Authors · 2025-08-15

We sincerely thank the reviewers for their constructive feedback and support, recognizing strengths of our work:

- **Well-motivated, novel design:** our method partitions the generative trajectory into temporal segments handled by specialized velocity blocks, effectively addressing the temporal heterogeneity of the generation process. (ySpd, ByUL, 5kd7, wHnp).
- **Computational efficiency**: our blockwise flow matching reduces inference cost compared to vanilla flow matching while preserving or improving quality, offering strengths in computation–quality trade-offs (ySpd, 5kd7, 6WUg).
- **Comprehensive validation**: extensive experiments, comparing with competitive benchmarks, systematic ablations, and comprehensive visualizations that robustly validate our method (ByUL, 6WUg, wHnp).
- **Clarity and reproducibility**: clear writing, strong figures, and detailed implementation that make the work easy to follow and reproduce (ySpd, ByUL, 5kd7, wHnp, 6WUg).

---

During the discussion, we further strengthened the paper by:

- Demonstrating scalability to **text-to-image generation** and **high-resolution generation** (MS-COCO, ImageNet 512×512).
- Showing seamless adaptation to **few-step generation**.
- Providing detailed **computational comparisons**, confirming inference efficiency and training scalability.
- Clarifying architecture design and **addressing discontinuity concerns** between temporal segments.

---

We are pleased that the rebuttal and the author-reviewer discussion have resolved the most of the concerns raised by the reviewers, and we appreciate the reviewers (ByUL, 5kd7, wHnp, 6WUg) who indicated that **they will increase the score** based on our clarifications and additional results. We believe that the strengthened experimental evidence, clearer explanations, and well-defined future directions further enhance the quality and impact of the paper.

We thank the committee once again for their time and consideration.

---

### Decision · Program_Chairs · 2025-09-17

**Decision:**

Accept (poster)

**Comment:**

This paper proposes Blockwise Flow Matching (BFM), a new framework for flow matching generative models. Instead of a single large network handling the entire generative trajectory, BFM partitions the trajectory into temporal segments, each handled by a specialized smaller “velocity block.” To enhance quality and efficiency, the authors also introduce two components: (1) Semantic Feature Guidance, which conditions velocity blocks on semantically rich pretrained representations (DINOv2), and (2) Feature Residual Approximation (FRA), which approximates semantic features to reduce inference cost. Extensive experiments on ImageNet 256×256 demonstrate substantial Pareto improvements in FID vs. FLOPs, with additional results on higher-resolution (512×512) and text-to-image (MS-COCO) benchmarks showing scalability and generality.

All five reviewers viewed the paper as technically solid, with contributions that improve both efficiency and sample quality in flow matching models. Several reviewers (ByUL, 5kd7, 6WUg, wHnp) explicitly raised their scores after rebuttal clarifications and new results. While concerns remain about increased parameters, clarity of exposition, and novelty relative to prior step-size/dynamic scheduling works, the rebuttal convincingly positioned BFM as a distinct and complementary architectural innovation, which supports the decision of acceptance.